# From Static Benchmarks to Dynamic Protocol: Agent-Centric Text Anomaly Detection for Evaluating LLM Reasoning

**Seungdong Yoa**[1] **Sanghyu Yoon**[1] **Suhee Yoon**[1] **Dongmin Kim**[1] **Ye Seul Sim**[1]
**Junhyun Lee**[2] **Woohyung Lim**[1]
[1]LG AI Research, [2]Korea University
{seungdong.yoa, sanghyu.yoon, suhee.yoon, dmkim, ysl.sim}@lgresearch.ai
ljhyun33@korea.ac.kr, w.lim@lgresearch.ai

## Abstract

The evaluation of large language models (LLMs) has predominantly relied on static datasets, which offer limited scalability and fail to capture the evolving reasoning capabilities of recent models. To overcome these limitations, we propose an agent-centric benchmarking paradigm that moves beyond static datasets by introducing a dynamic protocol in which autonomous agents iteratively generate, validate, and solve problems. Within this protocol, a teacher agent generates candidate problems, an orchestrator agent rigorously verifies their validity and guards against adversarial attacks, and a student agent attempts to solve the validated problems. An invalid problem is revised by the teacher agent until it passes validation. If the student correctly solves the problem, the orchestrator prompts the teacher to generate more challenging variants. Consequently, the benchmark scales in difficulty automatically as more capable agents are substituted into any role, enabling progressive evaluation of large language models without manually curated datasets. Adopting text anomaly detection as our primary evaluation format, which demands cross-sentence logical inference and resists pattern-matching shortcuts, we demonstrate that this protocol systematically exposes corner-case reasoning errors that conventional benchmarks fail to reveal. We further advocate evaluating systems along several complementary axes including cross-model pairwise performance and progress between the initial and orchestrator-finalized problems. By shifting the focus *from fixed datasets to dynamic protocols*, our approach offers a sustainable direction for evaluating ever-evolving language models and introduces a research agenda centered on the co-evolution of agent-centric benchmarks. We release our benchmark protocol, including code and data, at https://huggingface.co/datasets/LGAI-DILab/ATAD.

## 1 Introduction

Static benchmarks, such as MMLU [1], GSM8K [2] and Big-Bench [3], once served as reliable indicators of language model progress. However, frontier large language models (LLMs) now approach—or even surpass—human-level accuracy on many of these tasks [4, 5, 6]. Because these benchmark suites are finite, publicly accessible, and often included in pretraining corpora, models may inadvertently memorize substantial portions of the test data [7]. This can lead to inflated leaderboard results that do not reflect genuine improvements in reasoning ability. Unfortunately, it has become increasingly difficult to draw meaningful distinctions from these overused datasets. First, data contamination is now common: large-scale data collection often includes benchmark questions in pretraining datasets, and efforts to remove them afterward are usually incomplete [8]. Second,

Submitted to 39th Conference on Neural Information Processing Systems (NeurIPS 2025). Do not distribute.

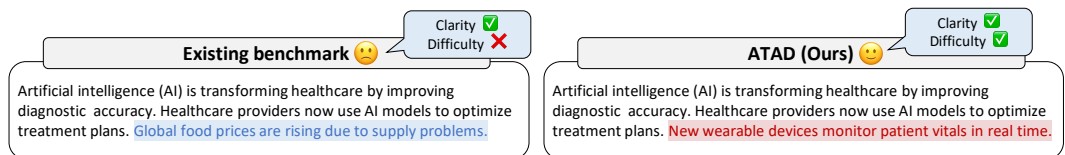

Figure 1: **Comparison of text anomaly samples.** *Left*: Existing benchmarks include obvious anomalies (e.g., complete off-topic from sports news to economy news) that are clear but too trivial. *Right*: ATAD examples introduce subtle shifts within context (e.g., benefits to ethics in healthcare AI), preserving clarity while presenting reasoning-intensive challenges. Our collaborative agents resolve the clarity-difficulty trade-off through iterative task refinement.

because static benchmarks contain a limited number of items, model developers may—sometimes without realizing it—tune their systems to match the details of these benchmarks. This creates feedback loops that improve scores without real gains in general reasoning ability [9]. Third, once a benchmark is considered "solved", the research community must quickly create a new one. This leads to a cycle of rapid creation and decline, which uses up valuable time and provides only short-term insight into model performance [10]. These limitations highlight the inherent shortcomings of static benchmarks in evaluating real reasoning capabilities.

To overcome these shortcomings, dynamic benchmarks for LLMs are essential as they continuously evolve, mitigating data contamination and preventing models from overfitting to finite test sets. In particular, text anomaly detection serves as a powerful task to reveal subtle reasoning flaws, providing clearer insight into the true capabilities and limitations of LLMs [11]. However, constructing high-quality text anomaly detection problems remains challenging: increasing the difficulty often sacrifices clarity, while ensuring clarity typically results in overly simple tasks. Figure 1 illustrates this trade-off and motivates our protocol's design. We introduce the **A**gent-centric **T**ext **A**nomaly **D**etection (**ATAD**), a benchmark protocol that replaces the static-dataset paradigm with a three-agent system. In this protocol, as illustrated in Figure 2, a teacher agent generates candidate problems, an orchestrator agent validates them and filters out defective items, and a student agent attempts to solve the qualified problems. As a problem format, reasoning-centric anomaly detection tasks are well suited for evaluating LLMs: they require cross-sentence logical inference, resist pattern-matching shortcuts and training data leakage, and support objective, fine-grained scoring. Asking a model to identify and explain the single sentence that disrupts a passage's coherence offers a precise and robust measure of reasoning ability—one that is less prone to exploitation than many existing benchmarks. By shifting the focus *from fixed datasets to dynamic protocols*, we offer a sustainable direction for evaluating ever-evolving language models and invite the community to explore a research agenda in which models and the benchmarks that probe them co-evolve. We release an open-source reference implementation with empirical results showing that ATAD surfaces reasoning weaknesses invisible to static benchmarks. A comprehensive discussion of related work on dynamic benchmarking and text anomaly detection is provided in Appendix.

## 2 ATAD: Benchmark Protocol Design and Operation

We introduce a novel agent-centric dynamic benchmarking protocol, Agent-Centric Text Anomaly Detection (ATAD), illustrated in Figure 2. ATAD is designed to construct an adaptive benchmark for text anomaly detection by leveraging a teacher-student competitive loop and an orchestrator-regulated validation mechanism. Unlike static datasets, our protocol dynamically evolves problem difficulty based on student model performance while ensuring clarity and fairness through rigorous validation. This design enables the benchmark to scale with the capabilities of emerging language models, supporting sustainable and progressively challenging evaluation over time.

### 2.1 Agent Roles

**Teacher Agent**: Generates problems and increases their difficulty when the Student solves them correctly, forming a competitive loop that adapts to the Student's capabilities.
**Orchestrator Agent**: Validates the generated problem to ensure it is well-formed, unambiguous, aligned with the expected task type, and free from adversarial design. It also checks whether the problem is logically coherent and appropriately matches the intended difficulty level.
**Student Agent**: Attempts to solve the validated problem. If it succeeds, the problem is made harder; if it fails, the problem is accepted into the benchmark.

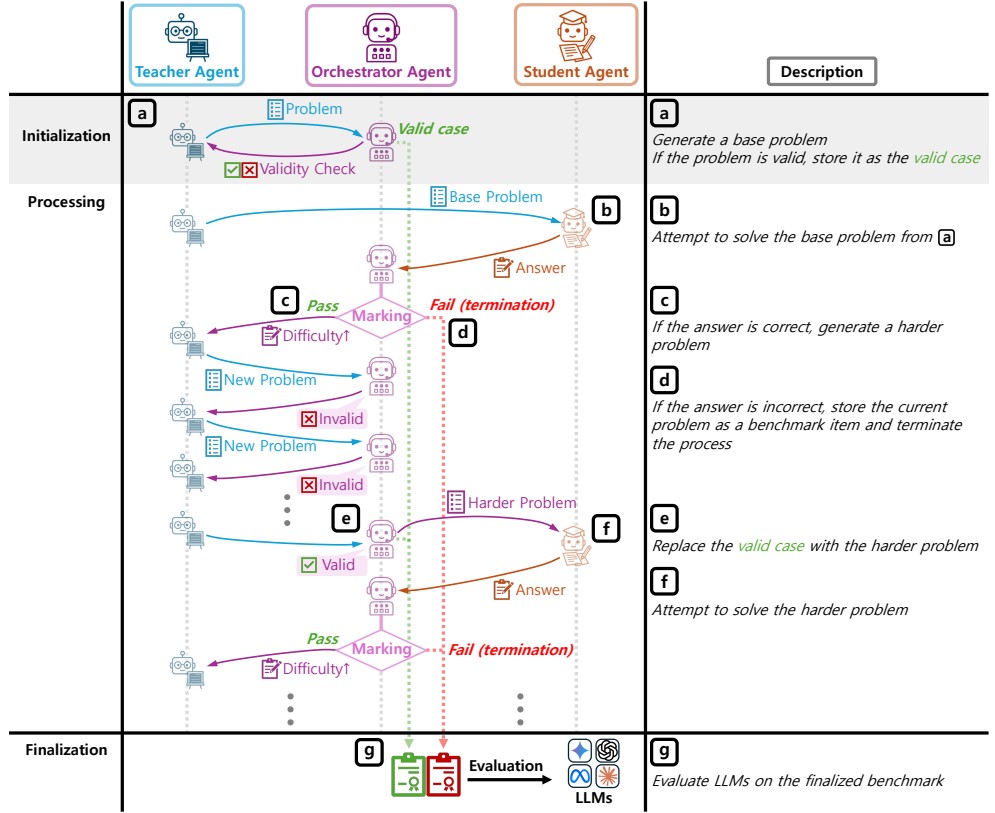

Figure 2: **Illustration of the overall ATAD protocol.** Three agents iteratively interact to generate progressively challenging benchmarks designed to uncover subtle reasoning weaknesses in LLMs.

The naming of Teacher and Student refers to agent roles in the protocol and is unrelated to model training paradigms such as knowledge distillation. In our framework, the competitive interaction between the Teacher and Student agents is leveraged to drive difficulty escalation in benchmark construction. This dynamic, however, can risk generating ambiguous or adversarial problems in the pursuit of harder samples. To mitigate this, the Orchestrator agent plays a crucial role in ensuring quality and fairness at each iteration. This validation process is particularly important for tasks like text anomaly detection, where subtle shifts in coherence, semantics, or phrasing can easily compromise problem clarity.

## 2.2 Protocol Phases

Our proposed benchmark construction protocol operates through a multi-agent system involving a Teacher, an Orchestrator, and a Student agent. These agents interact through two core phases: the Initialization Phase and the Adaptive Difficulty Scaling Phase. Each phase features automatic iteration control mediated by the Orchestrator. A visual summary of the protocol workflow is provided in Figure 2, with steps annotated from **a** to **g**.

### 2.2.1 Initialization Phase (Base Problem Generation)

The protocol begins with the Teacher agent generating a base-level problem for a designated text anomaly detection task (e.g., semantic deviation, sentence order inconsistency), corresponding to the **label a** in Figure 2. These base problems are intended to be of low difficulty and serve as the starting point for the benchmark construction.

Each generated problem is submitted to the Orchestrator for a multi-criteria validation process. The Orchestrator evaluates the sample for well-formedness, clarity, logical coherence, task type adherence, and fairness, while guarding against adversarial design or unanswerable ambiguity.

If the problem is invalid, the Orchestrator returns detailed feedback to the Teacher, prompting regeneration. This loop is governed by the Orchestrator's validation decisions and continues until a

valid problem is produced or a maximum number of attempts (`max_init_loops`) is reached. Once the problem passes validation, it is stored as a valid base problem and passed on to the Adaptive Difficulty Scaling Phase.

### 2.2.2 Adaptive Difficulty Scaling Phase

This phase begins with the Student's first attempt at the validated base problem and corresponds to the **b** through **f** labels in Figure 2. The Student attempts to solve the base problem (**label b**). If the Student fails, the problem is finalized as a benchmark item (**label d**), as it exposes a limitation in the Student's current reasoning capacity.

If the Student succeeds, the Orchestrator prompts the Teacher to generate a more challenging variant of the problem (**label c**). The Teacher, informed by the Student's prior success, creates a harder version aimed at pushing the Student's capabilities further. This new problem undergoes the same validation process by the Orchestrator to ensure that difficulty has increased meaningfully without compromising task clarity or fairness (**label e**).

Once validated, the harder problem replaces the previous one and is presented to the Student for another attempt (**label f**). This cycle—solving, regenerating, validating—continues iteratively until the Student fails or the iteration cap (`max_student_loops`) is reached. If the Teacher's harder problem is rejected by the Orchestrator, it may be prompted to slightly reduce the difficulty and regenerate, preserving the same task structure while avoiding ambiguity or excessive complexity. Although this does not constitute a formal decrease in the difficulty level, it allows for iterative refinement within the same hardness tier. If multiple regeneration attempts fail to produce a valid harder problem, the process terminates with the last previously validated problem—typically the one that the Student successfully solved—being finalized as the benchmark item.

The most difficult validated problem that causes the Student to fail is adopted as the finalized benchmark item. This structure allows the benchmark to automatically calibrate difficulty per instance, producing finely tuned evaluation samples based on actual model behavior.

### 2.2.3 Evaluation Phase

This phase corresponds to the **label g** in Figure 2. After benchmark samples are finalized through the above process, LLMs can be evaluated using the curated benchmark. Each problem is associated with its final difficulty level and validation metadata, supporting both overall performance comparisons and fine-grained reasoning diagnostics.

## 2.3 Key Features

Our benchmarking framework is grounded in two complementary principles: a competitive protocol in which the Teacher challenges the Student with progressively harder problems, and an adaptive validation mechanism where the Orchestrator ensures that difficulty scaling remains fair, coherent, and well-formed. Together, these two dynamics enable ATAD to produce reliable, high-quality benchmarks tailored to a model's actual reasoning capacity.

**Difficulty Scaling via Teacher-Student Competition.** The Teacher agent is implicitly incentivized to analyze the Student's prior successes and failures. This allows it to generate novel problems that directly target the Student's weaknesses or extend beyond its current competence, yielding more sophisticated samples than mere perturbations of existing items. Difficulty is adjusted dynamically based on the Student's performance, forming a competitive loop that drives benchmark depth.

**Orchestrator-Regulated Difficulty Control.** To prevent uncontrolled or adversarial difficulty escalation, the Orchestrator agent validates each problem before it is presented to the Student. It checks logical coherence, task adherence, clarity, and difficulty appropriateness, and autonomously decides whether the Teacher should regenerate a sample. This ensures that problem progression remains both challenging and fair, balancing the Teacher's incentives with principled quality control.

**Autonomous Iteration Control.** Unlike benchmarks with fixed iteration schedules, ATAD relies on the Orchestrator to dynamically determine when the Teacher should regenerate a problem or proceed to evaluation. This mechanism replaces manual tuning with agent-driven adaptability, ensuring high-quality, context-appropriate problems at every step.

**Failure-Driven Sample Finalization.** Problems are finalized not at creation, but at the point of

Student failure. This empirical approach anchors benchmark difficulty in actual model limitations rather than manual labels, surfacing failure cases that are often missed in static datasets.

**Dynamic Difficulty Localization.** Unlike benchmarks that assign difficulty globally, ATAD adjusts difficulty at the instance level based on Student feedback. This enables precise, localized probing of reasoning weaknesses and model-specific blind spots.

**Cross-Agent Instantiability.** ATAD is modular by design and supports different model pairings (e.g., $\text{ATAD}^{\text{gpt-4o}}_{\text{gemini2-flash}}$), enabling comparative evaluation and tracking of model evolution over time.

**Broad Task Coverage.** Our benchmark spans seven types of text anomaly detection tasks (see Section 3.2), capturing a wide range of reasoning capabilities including discourse coherence, contradiction detection, referential clarity, and stylistic consistency.

# 3 Task Design for Text Anomaly Detection

This section presents our design of text anomaly detection tasks as a probe of LLM reasoning (Section 3.1) and introduces a taxonomy of seven anomaly types (Section 3.2).

## 3.1 Task Overview and Motivation

We identify text anomaly detection as a particularly suitable domain for evaluating the reasoning capabilities of LLMs. These tasks target subtle inconsistencies in logic, coherence, or semantics, requiring genuine cross-sentence inference and resisting shortcuts based on surface-level patterns. However, creating high-quality text anomaly problems remains challenging: increasing task difficulty often introduces ambiguity, while prioritizing clarity can lead to trivial or shallow problems. This trade-off is especially pronounced in language-based tasks, where, unlike math or science, answers lack grounding in formal rules. Yet standardized exams like the GRE, GMAT, and LSAT show that natural language questions can still demand structured reasoning with clear answer standards. Inspired by these formats, our benchmark emphasizes deep reasoning while maintaining clarity and objectivity. Still, generating such problems at scale—especially in text anomaly detection—remains difficult, as it requires balancing subtlety and unambiguity. Our adaptive benchmarking protocol addresses this via a teacher-student competition regulated by an orchestrator, forming a self-calibrating system that reliably surfaces nuanced reasoning failures in LLMs.

## 3.2 Task Taxonomy: Seven Types of Text Anomalies and Reasoning Skills Targeted

Each task in our taxonomy is designed to assess a distinct aspect of LLM reasoning, such as coherence, logical consistency, or ambiguity resolution—areas often underrepresented in existing benchmarks. Together, the seven task types provide a broad and fine-grained evaluation of language understanding. While each task targets a core reasoning capability, we further diversify the benchmark by selectively incorporating anomaly factors known to challenge LLMs, including subtle semantic shifts or structural inconsistencies. These additions are applied to a subset of examples to enhance difficulty without sacrificing clarity or task diversity.

**T1. Sentence Context Anomaly** targets *contextual reasoning*, requiring the model to detect semantic inconsistencies between individual sentences and the paragraph's main theme. Challenge factors include *minor topic shifts* and *semantic deviations* that appear grammatically well-formed but subtly disrupt thematic coherence.

**T2. Paragraph Order Consistency** assesses *discourse coherence* by determining the correct order of sentences based on topic flow, causal and temporal dependencies. Challenge factors involve *sentence reordering* that appears locally coherent but requires comprehensive understanding of global document structure to detect.

**T3. Blank-based Choice Anomaly** requires both *lexical* and *pragmatic reasoning* to identify an inappropriate word or phrase within context. Challenge factors focus on *lexical fit* and *collocation*, requiring the detection of choices that are grammatically correct but contextually inappropriate. This demands both common sense and sensitivity to subtle nuances.

**T4. Bridge Sentence Evaluation** focuses on *logical bridging* and *topic shift detection*, requiring the model to judge whether a candidate sentence logically connects two related paragraphs. Challenge factors include *weak logical connections* and *abrupt topic shifts*, where the sentence itself may seem plausible but fails to maintain coherent discourse flow.

**T5. Referential Ambiguity** tests *coreference resolution* to identify sentences where pronouns or referring expressions are ambiguous or misleading, disrupting clarity in discourse interpretation. Challenge factors involve *ambiguous pronouns* and *unclear references* that disrupt sentence clarity.

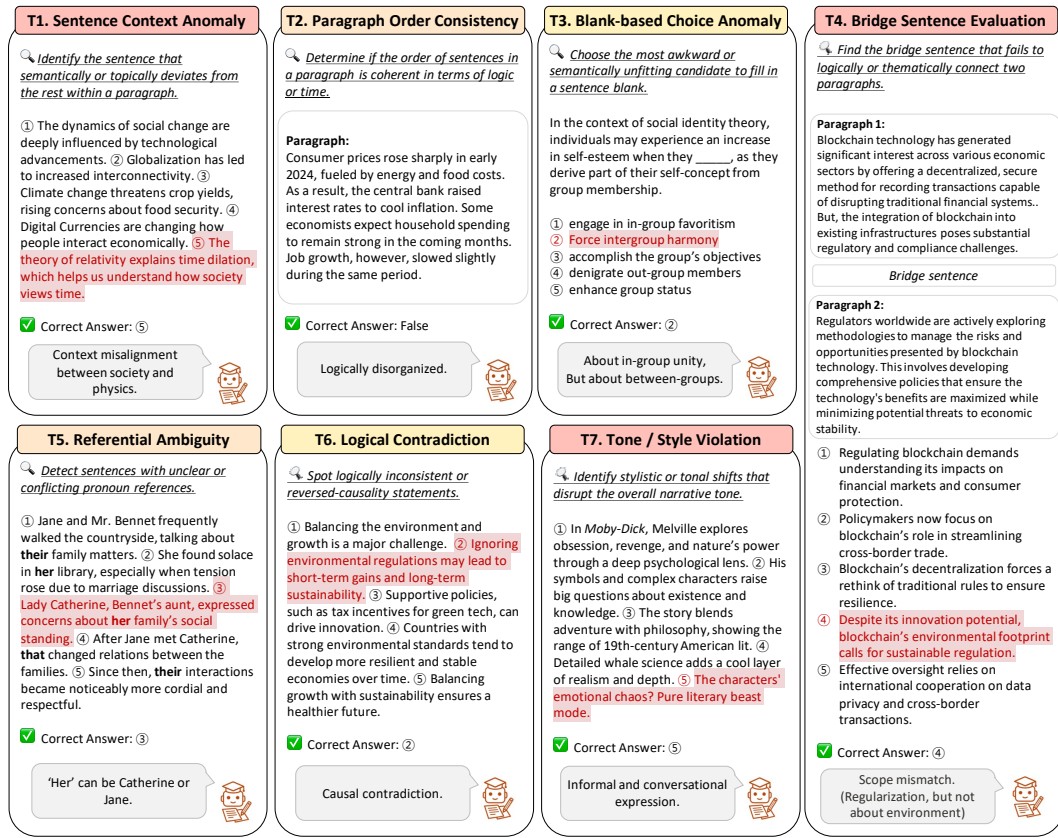

Figure 3: **Examples of the seven task types of text anomalies.**

**T6. Logical Contradiction** measures *causal and contradiction reasoning*. The model detects inconsistencies such as violated cause–effect relationships or misinterpreted correlations as causation. Challenge factors include *contradictory claims* and *causal reversals*.

**T7. Tone/Style Violation** evaluates *stylistic reasoning* by assessing whether all sentences maintain a consistent tone and register (e.g., formal vs. informal). The model must identify any sentence that deviates from the overall style. Challenge factors include *tone shifts* and *register mismatches* that subtly undermine stylistic coherence.

While each task focuses on a primary reasoning skill, practical cases often demand the integration of multiple capabilities—such as critical thinking and fine-grained semantic analysis. For example, T4 not only requires assessing logical coherence but also detailed semantic understanding. Building on this, we enhance task diversity by incorporating them within six academic domains frequently found in standardized reasoning exams (*e.g.,* GRE, LSAT), including science, philosophy, politics/society, psychology, economics, and literature. Rather than assigning domains randomly, we systematically align them to the tasks where the domain's inherent characteristics amplify reasoning challenges. This principled topic-to-task mapping is detailed in Appendix, with additional examples and domain-specific motivations. Figure 3 outlines representative task formats, with full design details available in Appendix as well.

# 4 Experiments and Results

This section presents our experimental evaluation of the benchmark generated through our protocol, highlighting its utility for assessing LLM reasoning. We evaluate overall performance, examine the Teacher-Student competition protocol for difficulty scaling, and assess the contribution of Orchestrator validation. Additionally, we explore its use in forecasting future LLM capabilities and test its consistency across multiple runs.

Table 1: Overall Performance of LLMs on our Text Anomaly Detection Benchmark. Average accuracy of each LLM, across the four datasets generated by four agent families (GPT, Gemini, Claude, LLaMA), is shown for each anomaly type (T1-T7) and overall.

| Evaluation Model | T1 | T2 | T3 | T4 | T5 | T6 | T7 | Avg. |
|---|---|---|---|---|---|---|---|---|
| GPT-3.5-Turbo [16] | 59.00 | 16.00 | 66.75 | 48.50 | 55.75 | 51.75 | 81.50 | 54.18 |
| GPT-4o-mini [17] | 57.25 | 17.00 | 62.50 | 54.00 | 52.50 | 58.75 | 83.00 | 55.00 |
| GPT-4o [12] | 62.00 | 21.25 | 68.25 | 53.25 | 49.25 | 56.75 | 81.00 | 55.96 |
| GPT-o4-mini [18] | 63.25 | 30.25 | **68.50** | 53.00 | 47.25 | 57.25 | 80.00 | 57.07 |
| Gemini-1.5-Flash [19] | 6.00 | 11.25 | 62.00 | 48.75 | 17.50 | 10.75 | 21.00 | 25.32 |
| Gemini-2.0-Flash-Lite [14] | 64.00 | 10.75 | 63.50 | 52.25 | **62.75** | **62.00** | 86.25 | 57.36 |
| Gemini-2.0-Flash [14] | 65.25 | 25.00 | 63.00 | 58.25 | 51.00 | **62.00** | **88.00** | 58.93 |
| Claude-3-Haiku [20] | 63.75 | 12.00 | 61.00 | 51.75 | 53.50 | 60.00 | 72.75 | 53.54 |
| Claude-3.5-Haiku [13] | 19.75 | **55.00** | 7.25 | 5.00 | 5.50 | 8.50 | 35.50 | 19.50 |
| Claude-3.5-Sonnet [13] | **65.75** | 31.75 | 65.00 | 59.50 | 53.50 | 57.50 | 86.75 | **59.96** |
| LLaMA-3.1-8B [15] | 39.50 | 12.75 | 35.50 | 24.50 | 53.00 | 38.75 | 68.75 | 38.96 |
| LLaMA-3.3-70B [15] | 60.75 | 27.75 | 63.25 | **60.00** | 52.25 | 57.75 | 84.25 | 58.00 |

## 4.1 Evaluation Setup

To evaluate LLM performance on our text anomaly benchmark, we established the following setup:

**Benchmark dataset.** The benchmark dataset comprises 700 samples per generation model, with 100 instances for each of the seven task types.

**Generation models.** We used the following LLMs as Teacher, Student, and Orchestrator agents to generate the benchmark datasets: GPT-4o [12], Claude-3.5-Sonnet [13], Gemini-2.0-Flash [14], and LLaMA-3.3-70B [15]. (When not explicitly stated, the Teacher, Student, and Orchestrator agents within a generation process use the same LLM.)

**Evaluation models.** We evaluated the generated datasets using a diverse set of LLMs: GPT-3.5-turbo, GPT-4o-mini, GPT-4o, GPT-o4-mini, Claude-3.0-Haiku, Claude-3.5-Haiku, Claude-3.5-Sonnet, Gemini-1.5-Flash, Gemini-2.0-Flash-Lite, and Gemini-2.0-Flash. These models serve as our baseline for assessing the difficulty and effectiveness of the benchmark.

## 4.2 Overall Performance Evaluation

Table 1 presents the overall performance of various LLMs on our text anomaly detection benchmark. We report accuracy as the primary evaluation metric, calculated as the proportion of correctly identified anomalies. Table 1 showcases the average accuracy achieved by each evaluation model across the four distinct benchmark datasets, each generated by a different agent family: GPT, Gemini, Claude, and LLaMA. For the benchmark generation process, the Teacher, Student, and Orchestrator agents were configured to be the same LLM for simplicity (e.g., GPT-4o for all three roles within the GPT-generated benchmark).

The results reveal a varied landscape of performance across different anomaly types (T1–T7). Notably, no single evaluation model consistently outperformed others across all categories, suggesting that the nature of the anomaly significantly influences detection accuracy. Claude-3.5-Sonnet achieved the highest overall average accuracy (59.96%), indicating strong general capability. However, other models surpassed Claude on specific types: GPT-4o-mini outperformed Claude on T3 by 3.5%, and Gemini-2.0-Flash exceeded Claude on T6 by 4.5%. Interestingly, certain evaluation models showed remarkable proficiency in specific anomaly types. Claude-3.5-Haiku, despite its relatively lower overall average (53.54%), achieved the highest accuracy in detecting anomalies of type T2 (55.00%). This highlights the potential for certain models to possess specialized strengths in identifying particular kinds of textual irregularities. While the overall average accuracy across all models and anomaly types indicates the inherent difficulty of the task, the varying performance across different anomaly types underscores the benchmark's ability to probe diverse aspects of LLM understanding and reasoning regarding text anomalies.

## 4.3 Valid Difficulty Scaling via Competitive Agents

To assess whether our competitive protocol effectively scales problem difficulty, we compare evaluation model performance on the initial base problems and the finalized benchmark versions. Table 2 presents the average accuracy of each evaluation model, computed across the seven anomaly types (T1–T7), on four benchmark datasets generated by different agent families. The Base datasets represent the initial set of generated problems before difficulty scaling, while the Final datasets are the

Table 2: Comparison of the LLMs' performance on the initial (base) datasets, consisting of the base problems, and the final versions of the benchmark datasets. Each column represents a different dataset, generated by GPT-4o, Claude-3.5-Sonnet, Gemini-2.0-Flash, and LLaMA-3.3-70B, respectively. The observed performance drop from base to final problems highlights the effectiveness of ATAD in exposing the weaknesses of LLM reasoning.

| Evaluation Model | GPT-4o | | Gemini-2.0-Flash | | Claude-3.5-Sonnet | | LLaMA-3.3-70B | |
|---|---|---|---|---|---|---|---|---|
| | Base | Final | Base | Final | Base | Final | Base | Final |
| GPT-3.5-turbo | 91.00 | 67.71 | 80.00 | 42.00 | 83.71 | 61.43 | 86.00 | 45.57 |
| GPT-4o-mini | 93.00 | 68.29 | 80.43 | 42.71 | 84.14 | 57.86 | 87.43 | 51.14 |
| GPT-4o | 94.29 | 72.43 | 83.29 | 44.71 | 87.29 | 62.71 | 89.29 | 44.00 |
| GPT-o4-mini | 91.86 | 72.43 | 83.57 | 47.14 | 87.29 | 61.86 | 87.71 | 46.86 |
| Gemini-1.5-Flash | 50.57 | 30.29 | 40.14 | 17.00 | 40.43 | 28.29 | 41.43 | 25.71 |
| Gemini-2.0-Flash-lite | 92.43 | 69.14 | 81.57 | 45.43 | 83.86 | 58.86 | 85.14 | 56.00 |
| Gemini-2.0-Flash | 92.29 | 71.86 | 82.43 | 44.29 | 85.43 | 61.86 | 88.00 | 57.71 |
| Claude-3-Haiku | 91.57 | 67.86 | 79.43 | 42.86 | 82.71 | 54.57 | 83.43 | 48.86 |
| Claude-3.5-Haiku | 36.86 | 18.86 | 39.86 | 24.71 | 39.71 | 18.57 | 45.86 | 15.86 |
| Claude-3.5-Sonnet | 91.71 | 72.86 | 83.86 | 47.43 | 88.86 | 63.29 | 88.29 | 56.29 |
| LLaMA-3.1-8B | 67.29 | 47.00 | 59.57 | 28.57 | 63.57 | 33.57 | 64.14 | 46.71 |
| LLaMA-3.3-70B | 93.43 | 72.43 | 82.71 | 43.57 | 89.29 | 64.57 | 92.43 | 51.43 |

Table 3: Comparison of LLMs' Performance and Problem Quality on the benchmark generated by GPT-4o agents. Problem quality is evaluated by each model acting as a reviewer, comparing benchmarks generated with and without the use of an Orchestrator.

| Evaluation Model | Performance (%) | | Problem Quality | | | | | | | |
|---|---|---|---|---|---|---|---|---|---|---|
| | w/o Orch. | w/ Orch. | Validity (1–5) | | Coherence (1–5) | | Fairness (1–5) | | Approval Rate (%) | |
| | | | w/o Orch. | w/ Orch. | w/o Orch. | w/ Orch. | w/o Orch. | w/ Orch. | w/o Orch. | w/ Orch. |
| GPT-4o | 68.29 | 72.43 | 4.30 | 4.85 | 3.71 | 4.74 | 3.20 | 4.65 | 38.14 | 87.14 |
| Gemini-2.0-Flash | 65.00 | 71.86 | 5.00 | 5.00 | 4.97 | 5.00 | 4.93 | 4.94 | 99.00 | 100.00 |
| Claude-3.5-Sonnet | 65.00 | 72.86 | 4.61 | 4.92 | 4.11 | 4.69 | 3.41 | 4.42 | 55.57 | 90.43 |
| LLaMA-3.3-70B | 65.71 | 72.43 | 4.66 | 4.87 | 4.37 | 4.76 | 4.34 | 4.80 | 66.00 | 88.29 |

result of the subsequent Teacher-Student competition and Orchestrator validation processes, designed to increase the benchmark's difficulty.

Across all agent families and evaluation models, we observe a consistent drop in accuracy from the base to final benchmarks. This indicates that our protocol successfully increases task difficulty in a controlled manner. On average, evaluation accuracy drops by approximately 37.3 percentage points after the adaptive scaling phase, highlighting the non-trivial nature of the final problems. Importantly, despite the increased difficulty, the final problems maintain high quality, as validated separately (see Section 4.4). This substantial reduction in accuracy confirms that the competitive interaction between the Teacher and Student agents, coupled with the Orchestrator's validation, successfully led to the creation of more challenging anomaly detection instances.

## 4.4 Orchestrator Validation

This section underscores the crucial role of the Orchestrator agent in ensuring the quality and validity of our text anomaly detection benchmark. To demonstrate this, we compared the performance of several LLMs on two versions of a benchmark generated by GPT-4o agents: one created solely through the Teacher-Student competition protocol (without an Orchestrator) and the other generated using our full framework, including Orchestrator validation. Table 3 presents this comparison, showing the evaluation performance of GPT-4o, Gemini-2.0-Flash, Claude-3.5-Sonnet, and LLaMA-3.3-70B on both benchmark versions.

At first glance, the benchmark generated without an Orchestrator appears more challenging—evaluation accuracy is consistently lower across all models. However, when we analyze problem quality along dimensions such as validity, coherence (logical consistency and type adherence), and fairness, we observe a notable degradation in quality. This suggests that the lower performance is not due to truly challenging reasoning tasks but rather to flawed or ambiguous question design. In other words, the competitive protocol without validation tends to inflate difficulty artificially by generating problems that are confusing or ill-posed.

By contrast, our Orchestrator-guided pipeline maintains higher quality across all metrics while still increasing difficulty. The Orchestrator filters out problems that are ill-formed, inconsistent, or lack a clear solution, ensuring that performance drops are reflective of genuine reasoning challenges—not annotation noise or design failures. These findings emphasize the critical role of the Orchestrator in producing challenging yet fair benchmarks, where performance gaps more accurately reflect model capability rather than dataset artifacts.

## 4.5 Scenario: Evaluating Future LLM Capabilities

To examine the sustainability of our benchmark under the rapid pace of LLM advancements, we simulate a future scenario where newer models outperform the current generation. Specifically, we assume GPT-4o as the generation model—serving as Teacher, Student, and Orchestrator—and evaluate the resulting benchmark using GPT-o3-mini and GPT-o4-mini, hypothetical successors representing future LLMs.

As shown in Table 4, all models—including the current GPT-4o—achieve near-ceiling accuracy on the base problems, highlighting the limitation

Table 4: Simulated future scenario with GPT-o3/o4-mini (future) vs. GPT-4o/4o-mini (current), showing sustained relative evaluation.

| Evaluation Model | GPT-4o | |
|---|---|---|
| | Base | Final |
| GPT-o3-mini | 93.71 | 72.14 |
| GPT-o4-mini | 91.86 | 72.43 |
| GPT-4o | 94.29 | 72.43 |
| GPT-4o-mini | 93.00 | 68.29 |

of static benchmark design. However, when evaluated on the final benchmark constructed through our difficulty-scaling protocol, performance drops substantially for all models. Notably, GPT-o3-mini and GPT-o4-mini score lower than GPT-4o, despite being assumed as future improvements.

This demonstrates that our benchmark not only scales difficulty in response to the generator's capability but also maintains long-term relevance. Unlike static benchmarks that saturate over time, our framework supports **relative evaluation**, where difficulty dynamically adapts to each generation model, allowing performance gaps between models to remain meaningful. Even as LLMs grow more powerful, our protocol preserves discriminative power—enabling robust comparison across models, regardless of when they are developed.

## 4.6 Consistency and Stability in Benchmark Generation

To ensure that our benchmark protocol supports not only adaptability but also **reliable reproducibility**, we evaluate the consistency of benchmark quality across repeated generations. In this experiment, we repeatedly generate benchmark datasets using the same agent configuration—Gemini-2.0-Flash as the Teacher, Student, and Orchestrator—and measure the performance of GPT-4o-mini, a representative model from a different family (GPT series), on these benchmarks. We generate 50 samples per task (350 in total) in the first round, then incrementally add 50 samples per task in each subsequent round, up to 1000 samples per task. For each round (50 to 1000 samples), we evaluate GPT-4o-mini on the corresponding benchmark and track its average accuracy across the seven anomaly detection tasks. Figure 4 plots model accuracy

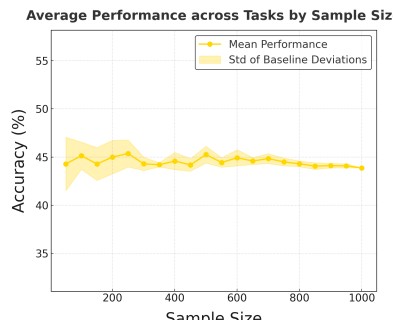

Figure 4: Consistency in Benchmark Generation.

per task as a function of the number of generated samples. We observe that performance remains largely stable across sample sizes, with only minor fluctuations. This result shows that our benchmark generation protocol is not only adaptive and dynamic, but also statistically stable across runs.

## 5 Conclusion

We present ATAD, an agent-centric benchmark protocol that adaptively generates and validates reasoning-focused anomaly detection tasks. By shifting from static datasets to dynamic protocols, ATAD enables sustainable, scalable, and stable evaluation of ever-evolving LLMs. Our results demonstrate that ATAD surfaces reasoning failures missed by conventional benchmarks and enables model-benchmark co-evolution, offering actionable insights into model-specific reasoning gaps. Future work includes extending ATAD to track evolving LLMs and advancing text anomaly detection as a reasoning benchmark.

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
