# Supplementary Material
# From Static Benchmarks to Dynamic Protocol: Agent-Centric Text Anomaly Detection for Evaluating LLM Reasoning

## A    Summary

In this supplementary material, we provide extended analyses and additional details to complement the main paper. Specifically, we include the following: (1) detailed performance results across benchmarks generated by four different models, expanding upon the averaged results in Table 1 of the main paper; (2) expanded descriptions of the seven anomaly detection task types, along with their associated topics and anomaly factors; (3) prompt templates and examples used for task generation and validation; (4) failure cases rejected by the Orchestrator during the benchmark validation phase; (5) performance details corresponding to Figure 4 of the main paper; (6) additional related work not included in the main text due to space constraints; and (7) future research directions for extending our protocol.

## B    Evaluation Results by Benchmark Generator

In Table 1, we report detailed accuracy tables for each benchmark individually generated by GPT-4o, Gemini-2.0-Flash, Claude-3.5-Sonnet, and LLaMA-3.3-70B, complementing the averaged results shown in Table 1 of the main paper. Each table presents the performance of the twelve evaluation models on a benchmark created by one of the four generator models.

Interestingly, as shown in Table 1, there is no single evaluation model that consistently outperforms others across all benchmarks. Although one might expect GPT-family models to perform best on the benchmark generated by GPT-4o, we observe that Claude-3.5-Sonnet achieves the highest average score in that case, as reported in Table 1a. This suggests that the identity of the generator model does not systematically favor or disadvantage any particular evaluation model family. The observed performance differences are more attributable to the inherent difficulty and heterogeneity of the benchmarks, rather than to any systematic advantage conferred to evaluation models by alignment with the generator.

## C    Descriptions of Anomaly Detection Task Types

This section provides additional details on the seven text anomaly detection tasks introduced in Section 3 of the main paper. Each task is designed to evaluate a distinct aspect of LLM reasoning, ranging from contextual and discourse coherence to ambiguity resolution and logical consistency.

As summarized in Table 2, we present the input and output formats for each task, reflecting how anomaly instances are structured and what form of prediction is expected from the model. These formats fall into three structural categories: (1) identifying an anomalous sentence within a paragraph (T1, T5, T6, T7), (2) selecting an inappropriate option from a given list of candidates (T3, T4), and (3) determining whether the overall sentence order in a paragraph is coherent (T2). The corresponding outputs are represented either as index selections or binary judgments, depending on the task type.

Beyond structural design, Table 3 outlines the core reasoning types targeted by each task, the specific challenge factors incorporated to enhance difficulty, and the domain topics used to amplify reasoning complexity. Challenge factors—such as subtle semantic deviations, logical reversals, or ambiguous pronouns—are selectively added to a subset of samples to increase difficulty while preserving clarity.

Table 1: Performance of evaluation models on benchmarks generated by GPT-4o, Gemini-2.0-Flash, Claude-3.5-Sonnet, and LLaMA-3.3-70B.

(a) Generated by GPT-4o

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

Table 2: Input/output structure for each text anomaly detection task.

| Task ID | Task Name | Input | Output |
|---|---|---|---|
| T1 | Sentence Context Anomaly | 5–6 sentence paragraph | Index of off-topic sentence |
| T2 | Paragraph Order Consistency | 5-sentence paragraph | Boolean (True/False) |
| T3 | Blank-based Choice Anomaly | Sentence with blank + 5 choices | Index of most inappropriate choice |
| T4 | Bridge Sentence Evaluation | Two paragraphs + 5 bridge candidates | Index of incoherent bridge |
| T5 | Referential Ambiguity | 5-sentence paragraph | Index of ambiguous sentence |
| T6 | Logical Contradiction | 5-sentence paragraph | Index of logically inconsistent sentence |
| T7 | Tone / Style Violation | 5-sentence paragraph | Index of tone/style violation |

Table 3: Reasoning types, challenge factors, and domain topics per anomaly detection task.

| Task ID | Reasoning Type | Challenge Factors | Topics (Domains) |
|---|---|---|---|
| T1 | Contextual reasoning | Minor topic shift, semantic deviation | Philosophy, society, psychology |
| T2 | Discourse coherence | Sentence reordering | Science, economics, politics |
| T3 | Lexical + pragmatic reasoning | Lexical fit, collocation | Literature, psychology, philosophy |
| T4 | Logical bridging + topic shift detection | Weak logical connection, abrupt topic shift | Economics, society, policy |
| T5 | Coreference resolution | Ambiguous pronouns, unclear referents | Psychology, literature, philosophy |
| T6 | Causal and contradiction reasoning | Contradictory claims, causal reversal | Science, economics, politics |
| T7 | Stylistic reasoning | Tone shift, register mismatch | Literature, philosophy |

To promote diversity and prevent overfitting to specific patterns, each factor is applied with a 50% probability during problem generation.

To ensure comprehensive coverage of academic reasoning, task content is curated across six high-level domains (e.g., science, economics, philosophy). Each task is paired with domains that naturally emphasize the relevant reasoning challenge, facilitating a principled topic-to-task alignment. This mapping is shown in the final column of Table 3.

While each task is designed around a primary reasoning capability, many demand compound reasoning skills—for example, Task T4 (Bridge Sentence Evaluation) requires not only logical coherence but also sensitivity to topic transitions. Such multifaceted design enables our benchmark to assess nuanced reasoning failures beyond surface-level understanding.

# D  Prompt Templates and Examples

This section presents the prompt templates used in our benchmark pipeline. We categorize prompts into two primary roles: (1) generation prompts used by the **Teacher agent** to construct task instances, and (2) validation prompts used by the **Orchestrator agent** to assess the quality and structure of those instances.

The generation prompts are designed to be style-specific (e.g., GRE-style) and conditionally incorporate difficulty scaling instructions and challenge factors (e.g., semantic deviation, logical inconsistency). Each prompt guides the Teacher to generate one of the seven anomaly task types (T1–T7) in a consistent JSON schema.

The validation prompts ensure that the generated problems are well-formed, solvable, and coherent. These prompts are used by the Orchestrator during three key phases of the protocol: (1) immediately after the Teacher generates an initial problem (Initial Validation); (2) after the Student solves the problem correctly, to provide feedback for generating a harder version (Feedback for Difficulty Escalation); (3) once a new, difficulty-scaled version of the problem is created, to ensure it maintains quality and appropriate challenge (Validation of Difficulty-Scaled Problem). Below, we present representative prompt examples for Task T1 across all three phases.

For clarity, we simplify the full prompt, and the full set of prompt templates used for these phases is publicly available on our GitHub and Hugging Face repositories.[1]

---

[1] GitHub: https://github.com/seungdongy/atad
Hugging Face: https://huggingface.co/datasets/LGAI-DILab/ATAD

## D.1 Teacher Prompt for Task T1 (Generation)

> You are a GRE-style exam question generator. Create a question for task T1 on the topic of psychology.
> Generate 5 to 6 sentences on psychology. One of them should be anomalous (e.g., semantically inconsistent or conceptually off-topic).
> The anomaly should be based on: semantic deviation.
> Create a non-trivial anomaly that requires careful reading to detect. It should be noticeable but not immediately obvious.
>
> Return the result strictly in JSON format:
> ```
>     {
>        "context": ["..."],
>        "anomaly_index": <integer>,
>        "meta": {
>             "source": "GRE",
>             "topic": "psychology",
>             "anomaly_type": "semantic deviation"
>        }
>     }
> ```

## D.2 Orchestrator Prompt for Task T1 (Validation)

### D.2.1 Initial Validation Prompt

> You are a benchmark quality controller evaluating if this problem is well-formed and structured correctly for task T1.
>
> Task Type: Sentence Context Anomaly (T1)
>
> Task Description: This task requires generating 5–6 sentences on a topic where one of them is anomalous (semantically inconsistent or conceptually off-topic). The anomaly should be detectable but not overly obvious, requiring careful reading to identify.
>
> Expected Structure:
> - "context": array of 5–6 sentences
> - "anomaly_index": integer indicating the anomalous sentence
> - "meta": source, topic, anomaly_type
>
> Context:
> 1. Sentence A
> 2. Sentence B
> ...
> 5. Sentence E
>
> Correct Answer: Option 4
>
> Evaluate the problem based on these criteria:
> 1. VALIDITY: Is the problem well-formed and complete?
> 2. TYPE ADHERENCE: Does the problem follow the expected task type requirements?
> 3. LOGICAL COHERENCE: Is the anomaly identifiable?
> 4. FAIRNESS: Is the problem fair and reasonable? Does it have a clear, unambiguous solution?
>
> Return your evaluation in JSON format:
> ```
>     {
>        "approved": boolean (true if the problem passes all criteria, false otherwise),
>        "feedback": null if approved, or detailed feedback if rejected addressing:
>                 - Problem construction issues
>                 - Anomaly ambiguity concerns
>                 - Specific improvement suggestions
>     }
> ```

### D.2.2 Feedback Prompt (Orchestrator to Teacher) for Difficulty Escalation

When a Student successfully solves a task, the Orchestrator analyzes the Student's explanation and provides structured feedback to help the Teacher generate a harder version of the problem. This feedback prompt includes the original problem, the student's reasoning, and a checklist to guide difficulty escalation. Below is the full prompt used for this purpose.

You are helping to create a harder version of a problem that a student has correctly solved. Analyze the student's solution and provide feedback.

Task Type: Sentence Context Anomaly (T1)
Current Difficulty: easy

ORIGINAL PROBLEM:
```
{
    "context": [
        "Cognitive dissonance occurs when individuals experience conflicting beliefs.",
        "It can cause discomfort and lead to attitude change.",
        "Festinger's theory explains how people resolve dissonance.",
        "Photosynthesis is the process by which plants convert light into energy.",
        "Dissonance reduction strategies include rationalization and denial."
    ],
    "anomaly_index": 3,
    "meta": {
        "source": "GRE",
        "topic": "psychology",
        "anomaly_type": "semantic deviation",
        "difficulty": "easy"
    }
}
```

Student's Explanation: "The sentence about photosynthesis is unrelated to the other sentences on cognitive dissonance. It's a semantic outlier."

Based on how the student solved this problem, provide feedback to create a more challenging version:
1. What aspects did the student easily identify?
2. How could the problem be made more subtle or complex?
3. Give specific suggestions for increasing difficulty.

Return your feedback in JSON format:
```
{
    "analysis": "Brief analysis of student solution",
    "suggestions": ["Specific suggestion 1", "Specific suggestion 2", ...],
    "difficulty_increase": "Summary of how to increase difficulty"
}
```

### D.2.3 Validation of Difficulty-Scaled Problem

After the Teacher generates a more difficult version of the original problem, the Orchestrator evaluates its quality to determine whether the sample meets the necessary criteria for inclusion in the benchmark. This prompt includes a task-specific description, the expected output structure, the difficulty level, and the sample content. The Orchestrator then assesses whether the problem is well-formed, challenging, and coherent. Below is the full prompt used in this validation phase for Task T1.

---

You are a benchmark quality controller evaluating if a problem with increased difficulty is well-formed and appropriate for task T1.

Task Type: Sentence Context Anomaly (T1)
Difficulty Level: hard

Task Description: This task requires generating 5–6 sentences on a topic where one of them is anomalous (semantically inconsistent or conceptually off-topic). The anomaly should be detectable but not overly obvious, requiring careful reading to identify.

Expected Structure: The expected JSON structure should include 'context' (array of 5-6 sentences), 'anomaly_index' (integer indicating which sentence is anomalous), and 'meta' (with source, topic, and anomaly_type).

Context:
1. Cognitive dissonance occurs when individuals experience conflicting beliefs.
2. It can cause discomfort and lead to attitude change.
3. Festinger's theory explains how people resolve dissonance.
4. Social conformity often influences decision-making in groups.
5. Dissonance reduction strategies include rationalization and denial.
6. A dissonance-free state enhances psychological consistency.

Correct Answer: Option 4

Note: While maintaining quality standards, be lenient in your evaluation. Accept problems that are reasonable and solvable, even if they have minor imperfections.

Evaluate the problem based on these criteria:
1. VALIDITY: Is the problem well-formed and complete?
2. TYPE ADHERENCE: Does the problem follow the expected task type requirements?
3. LOGICAL COHERENCE: Is the correct answer clearly identifiable?
4. FAIRNESS: Is the problem fair and reasonable? Does it have a clear, unambiguous solution?
5. DIFFICULTY: Is the difficulty appropriate for hard level?

Return your evaluation in JSON format:
    {
        "approved": boolean (true if the problem passes all criteria, false otherwise),
        "feedback": null if approved, or detailed feedback if rejected addressing:
                    - Problem construction issues
                    - Anomaly ambiguity concerns
                    - Difficulty appropriateness
                    - Specific improvement suggestions
    }

---

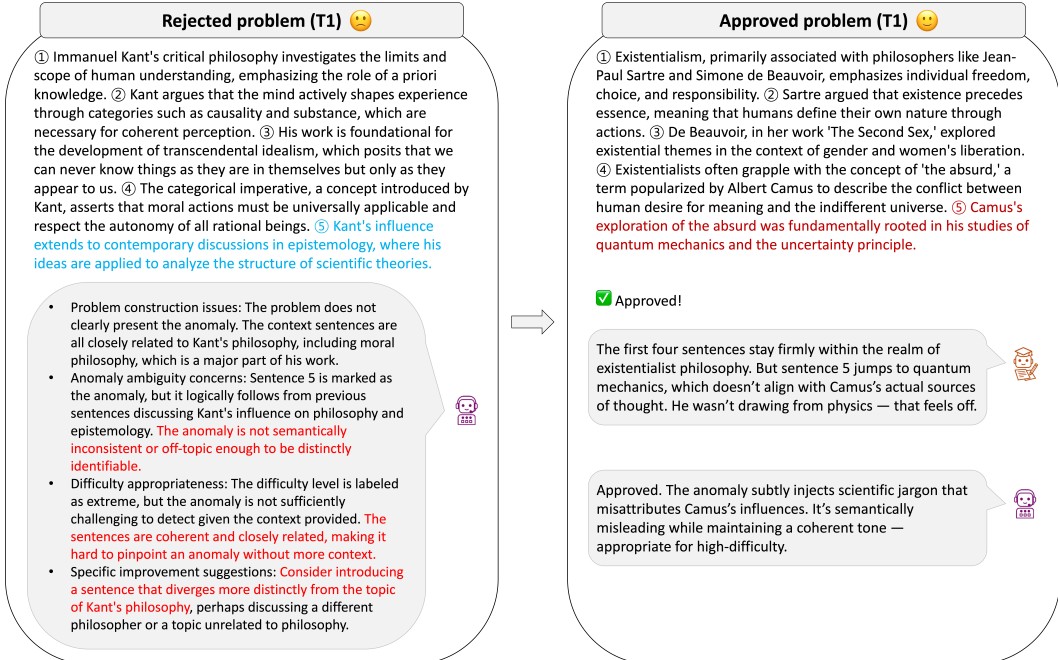

Figure 1: **Refined after rejection.** The left side shows a rejected T1 (Sentence Context Anomaly) problem where the anomaly was conceptually weak and difficult to identify. The Orchestrator's feedback noted the lack of semantic inconsistency and suggested stronger topic divergence. The revised version (right) introduces a scientifically framed yet incorrect statement about Camus's influences, resulting in a clearer and more pedagogically effective anomaly. This highlights the Orchestrator's role in guiding high-difficulty problem construction.

# E    Rejected Cases from the Orchestrator Validation

We present two distinct analyses to illustrate the role of the Orchestrator in problem validation. Section E.1 examines how rejections lead to improved samples by comparing rejected and subsequently approved versions. Section E.2 explores the consequences of using format-compliant but semantically flawed problems that were rejected, showing how such issues can affect model performance when the Orchestrator is removed.

## E.1    Refined After Rejection

One of the key functions of the Orchestrator is not just to detect flawed problems but to guide their improvement. Figure 1 illustrates a representative case from the T1 task (Sentence Context Anomaly), where the problem was rejected for lacking a clear anomaly and subsequently revised into a higher-quality version.

In the rejected version (left), all five sentences are factually correct and topically coherent, making it difficult to identify a distinct anomaly. The fifth sentence about Kant's influence on epistemology, while slightly tangential, remains within the bounds of acceptable variation in context. The Orchestrator flagged this problem as ill-suited for high-difficulty evaluation due to the lack of a clear semantic deviation.

After feedback, the Teacher produced a revised version (right) on existentialist philosophy, where the anomaly subtly introduces scientifically framed misinformation: it claims Camus's absurdism was based on quantum mechanics, which is factually incorrect. This revised problem is more appropriate for an "extreme" difficulty level as it requires nuanced understanding of philosophical context to detect the inconsistency.

This example demonstrates how Orchestrator feedback can elevate problem quality by transforming ambiguous or unfocused items into more challenging and pedagogically valid benchmark samples.

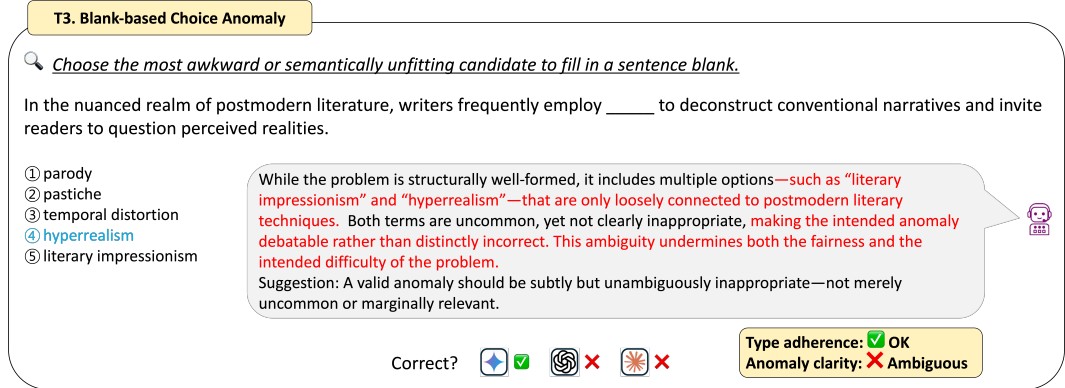

Figure 2: A structurally valid T3 problem rejected by the Orchestrator due to an unclear anomaly. Despite adhering to the task format, the anomaly is too ambiguous—leading two out of three LLMs to answer incorrectly.

## E.2  Structurally Valid but Semantically Flawed

This example highlights the importance of Orchestrator validation even when the problem format adheres to the expected task type. While structural correctness (e.g., number of options, sentence layout) can be verified without the Orchestrator, semantic soundness often cannot.

The problem shown here follows the T3 task format correctly but was rejected due to an unclear anomaly. Both "literary impressionism" and "hyperrealism" are loosely connected to the context, making the intended anomaly debatable.

Without validation, such problems could remain in the benchmark and appear reliable—yet when tested on three strong LLMs, two of them (GPT-4o and Claude 3.5 Sonnet) failed to answer correctly. This illustrates that structurally valid but semantically underspecified problems can mislead evaluation outcomes.

While this can happen in any phase, it becomes especially risky during difficulty escalation, where the Teacher agent may try to make problems harder but instead make them ambiguous.

## F  Performance Details for Consistency Figure

To support the results presented in Figure 4 of the main paper, this section provides additional details on how the consistency plot was computed. The figure tracks the average accuracy of GPT-4o-mini across different sample sizes—ranging from 50 to 1000 samples per task (T1–T7).

For each sample size, we evaluate the model's accuracy per task and calculate the average across tasks. The shaded region around the curve represents the standard deviation of task-wise deviations from the final round (i.e., the 1000-sample benchmark). For each sample size, we measure how much each task's accuracy differs from its corresponding value at 1000 samples, and compute the standard deviation across these deviations. This provides a straightforward view of consistency over time, showing that performance remains stable across all sample sizes—not just at the final stage.

As shown in the figure, the accuracy remains largely stable with only minor variations, indicating the robustness and consistency of our benchmark generation process.

## G  Related work

### G.1  Text Anomaly Detection Benchmarks

Recent benchmarks have begun explicitly evaluating an LLM's ability to detect linguistic anomalies and coherence breaks in text. For example, CoheSentia [10] introduces a human-annotated coherence dataset of 500 AI-generated paragraphs, with both holistic and incremental (sentence-by-sentence) coherence scores. DECOR [11] focuses on incoherence in L2 English writing, providing expert-

labeled context–sentence pairs for detecting coherence breaks, explaining their causes (e.g. lack of cohesion or consistency), and even rewriting the incoherent sentences. Disco-Bench [12] targets discourse-level anomalies by evaluating model performance on document-level test suites rich in cohesion and coherence phenomena across multiple tasks. Other well-known "anomaly" challenges include the Adversarial NLI dataset (ANLI)[13], collected in three rounds of human-and-model-in-the-loop adversarial examples, and the Winograd Schema Challenge[14] for commonsense pronoun disambiguation. These benchmarks share a focus on uncovering subtle inconsistencies or incoherence in text. However, they are inherently limited by their static, human-curated nature. Each new example often requires costly human creativity and annotation, making it difficult to sustainably scale up the dataset or progressively increase task difficulty. ANLI, for instance, achieved increasing complexity over three rounds, but this required extensive human involvement at each round. In general, static anomaly datasets incur high labeling costs and quickly saturate—once models learn to solve the fixed set of examples, evaluation stagnates, and creating harder cases demands significant manual effort. This motivates exploration of more dynamic and automated evaluation protocols for textual coherence and anomaly detection.

## G.2    Static LLM Evaluation Benchmarks

The standard paradigm for evaluating LLMs has been through fixed benchmarks covering a wide range of tasks. Notable examples include MMLU (Massive Multitask Language Understanding)[15], a 57-task exam covering diverse knowledge domains, GSM8K for grade-school math word problems[16], and BIG-Bench [17], a crowd-sourced collection of over 200 tasks probing various aspects of intelligence. These static benchmarks were initially effective for comparing models, but top-tier LLMs have rapidly saturated many of them. Models like GPT-4 now exceed or approach human-level performance on MMLU and GSM8K, leaving little headroom for differentiation. Moreover, concerns have arisen about *training data contamination*: since the evaluation sets are public and relatively small, powerful LLMs often inadvertently memorize or see similar questions during pre-training. This can inflate their scores without reflecting true reasoning progress, as evidenced by significant performance drops on rephrased or decontaminated test samples [18]. In short, static benchmarks are increasingly subject to memorization and ceiling effects. They also struggle to track evolving capabilities—once a benchmark is "solved" by current models, it cannot capture further improvements or new emergent reasoning skills. Beyond these, other important static benchmarks exist, such as AgentBench [19], VisualAgentBench [20], GAIA [21], ToolBench [22], and HumanEval [23], which primarily focus on isolated reasoning and generation capabilities of single agents, thereby failing to capture the intrinsic dynamics of multi-agent interactions. Recent efforts have proposed ever harder test sets (e.g. "MMLU 2.0" variants) and meticulous data filtering to mitigate leakage, but these are stopgap solutions. The inability of static evaluations to adapt alongside model progress motivates developing dynamic benchmarks that can continually pose fresh, unsolved challenges.

## G.3    Dynamic Benchmarks without Agents

A growing line of work aims to generate evaluation data *dynamically* – creating new test samples on the fly to match a model's ability – without relying on multi-agent interactions. DyVal [24] pioneered this approach with a general framework to algorithmically spawn new reasoning problems of controlled complexity. In DyVal, instead of a fixed dataset, a generation function produces test samples and a constraint mechanism modulates their complexity and validity in real time. One instantiation uses directed acyclic graphs to compose simple components into increasingly complex problems (e.g. multi-step math or logic puzzles), allowing systematic scaling of difficulty[24]. These graph-based generated tasks require genuine reasoning and cannot be solved by mere memorization, but DyVal's template-driven nature limits it to certain domains (math, logical puzzles, algorithms). DARG (Dynamic LLM Evaluation via Adaptive Reasoning Graph)[25] extends this idea by extracting the underlying reasoning graph of an existing benchmark problem and perturbing it to generate novel but related test samples. This yields new questions with tunable complexity levels while preserving coherence with the original data distribution[25]. Crucially, DARG uses an automated verifier (a code-augmented LLM) to ensure each generated sample's label or answer remains correct after perturbation, providing stronger ground truth guarantees. Broadly, these non-agent dynamic benchmarks demonstrate the ability to continuously adjust task difficulty and mitigate data contamination. Their main limitations lie in generality and validation: methods like DyVal rely on hand-crafted generation schemas (e.g. DAG operations) that are task-specific, while purely LLM-based generators risk

producing invalid or trivial questions without additional checking. Ensuring robust quality control often requires an auxiliary procedure (such as DARG's code executor or heuristic filters) given the absence of a human or agent "referee." Thus, dynamic sample generation has shown promise in maintaining evaluation challenge, but incorporating more general and trustworthy validation remains an open challenge.

## G.4 Agent-Based Dynamic Evaluation Frameworks

Recent approaches have started to leverage AI *agents* (LLMs themselves) to both generate new evaluation items and verify their quality, yielding self-refreshing benchmarks. These can be grouped by the role agents play:

### G.4.1 Agent-Based Problem Verification

Several frameworks employ one or more LLM agents as internal *judges* or *verifiers* to ensure evaluation data quality and correctness. Benchmark Self-Evolving [26] is a multi-agent system that iteratively refines existing benchmark questions: one agent perturbs the context or question (e.g. paraphrasing, adding noise or constraints) to make a new test instance, and another agent (or the model itself) attempts to solve it to verify that the instance is valid and non-trivial. By applying a set of such automated "reframing" operations, the benchmark can evolve dynamically with minimal human input. JudgeLM [27] demonstrates that an LLM fine-tuned as a specialized evaluator can reliably score or check open-ended answers with >90% agreement to human judgment, effectively serving as a scalable replacement for human evaluation. This idea of an LLM-as-judge is also used in many dynamic benchmarks to replace costly human verification: for example, DyVal's follow-up work introduces "meta-probing agents" that automatically generate and check new reasoning challenges [28], and the DARG framework's pipeline employs a code-execution agent to validate each generated sample's solution [25]. The BenchAgents system [29] goes even further in modularizing the process: it deploys separate LLM agents for planning what data to create, for actually generating candidate problems, for verifying the correctness/quality of each candidate, and finally for assembling the evaluation and scoring models on it. By having agents explicitly double-check answers or filter out flawed questions, these frameworks instill a degree of robustness into dynamically created benchmarks. The verification agents can catch mistakes or ambiguities that a single-pass generation might miss, ensuring that the evolving evaluation data remains challenging yet fair. A downside, however, is that the agents themselves (being imperfect LLMs) might introduce their own biases or occasional errors in judgment, so careful design and calibration of the "judge" agents is required to maintain reliability. Beyond general LLM evaluation, specific frameworks like PersonaGym [30] have emerged for assessing specialized agents, introducing the first dynamic evaluation framework and automated human-aligned metric (PersonaScore) for persona agents, which are LLM agents designed to act according to an assigned persona.

### G.4.2 Problem Generation via Multi-Agent Protocols

Other works explore multi-agent *interaction protocols*—such as collaboration, debate, or competition— to automatically generate or evaluate content in novel ways. ChatEval [31] is a representative example where multiple LLM-based critics form a "referee team" that debates and deliberates on the quality of a model's answer. By pitting several AI evaluators with different perspectives against each other in discussion, the evaluation becomes more robust than a single model's score, and the process mimics how multiple human judges arrive at a consensus. This multi-agent debate approach focuses on jointly evaluating content rather than generating new problems, but it showcases how agent interactions can replace and even surpass traditional human evaluation. In terms of content *generation*, BenchAgents (mentioned above) explicitly uses agents that cooperate (with minimal human oversight) to produce entirely new benchmark datasets — effectively automating the benchmark creation process through agent teamwork[29]. There are also emerging benchmarks to test the capabilities of multi-agent systems themselves. MultiAgentBench[32] evaluates how well LLM agents can collaborate or compete in shared environments and tasks, introducing scenarios where multiple agents communicate to solve a problem. Its contribution is a suite of multi-agent challenge tasks (with coordination protocols like star or graph networks and metrics for teamwork quality), rather than an evolving benchmark, but it underscores the interest in agent-agent interaction. Meeseeks[33] takes an iterative multi-turn approach: it simulates a realistic user interacting with an LLM by providing feedback on

failed requirements, and measures whether the LLM can self-correct over multiple rounds. While not explicitly framed as multi-agent (the "user" feedback could be programmatic), it creates a feedback loop akin to two agents – one posing and refining the request, the other improving its answers – working together to achieve a correct solution. Many of these multi-agent or multi-turn frameworks successfully generate complex, rich interactions, but notably, most lack an explicit adversarial or difficulty-raising dynamic. Agents often cooperate to improve quality (as in collaborative problem solving or debate), rather than engaging in competitive play where one tries to stump or outpace the other. Likewise, the tasks or evaluations are usually predefined or randomly sampled rather than *progressively ramped up* in response to a model's mastery. For example, ChatEval's agents are not trying to make the task harder – they are jointly judging a given response. MultiAgentBench provides diverse scenarios but does not adapt scenario difficulty based on performance. In short, current multi-agent evaluation protocols focus on novel ways to assess or create content (often leveraging the wisdom of multiple AI judges or creators), yet they stop short of introducing a competitive teacher–student dynamic or automated curriculum that continuously pushes the model to its limit.

# H   Future Works

## H.1   Game-Theoretic Formalization of the ATAD Protocol

A promising avenue for extending ATAD is to cast the Teacher–Orchestrator–Student loop itself as a cooperative game in which each agent's *move* (problem generation, validation, or solution) becomes a unit of experience whose marginal contribution to overall benchmark quality can be quantified with game-theoretic tools such as Shapley values and their ordered extensions — e.g., the Nowak-Radzik value that explicitly respects the temporal ordering of curriculum steps [34]. By treating successive rounds of ATAD as a sequence of coalitions, we could estimate how much each agent (or even each prompt strategy) accelerates difficulty calibration, then allocate compute or interaction budget proportionally to those cooperative gains; conversely, negative pairwise interactions would signal adversarial curricula or redundant checks that should be pruned. Embedding this credit-assignment mechanism inside the protocol would let ATAD adapt not only the problems it poses but also the roles and incentives of its constituent agents, yielding a self-tuning, game-theoretic benchmark that co-evolves with frontier LLMs while remaining transparent and fair. This transposition explicitly links ATAD's adaptive benchmarking to the proven game-theoretic curriculum framework [34], supplying both theoretical grounding and practical guidance for future protocol optimization.

## H.2   Meta-Agent Extensions to the ATAD Protocol

Recent advances in *meta agents*—agents that search over the design space of other agents—offer a promising path toward making ATAD self-improving, safer, and more sample-efficient. A meta agent that *programs new agents in code*—as in Meta Agent Search [35]—can iteratively discover superior teachers (richer problem generators) and orchestrators (sharper validators) while archiving each discovery for reuse. Complementarily, AFLOW's Monte-Carlo-Tree-Search over code-represented workflows can refine validation pipelines and student curricula so that even smaller models, paired with optimized workflows, rival larger baselines at a fraction of the compute cost [36]. Casting *"make the problem just hard enough"*, *"catch adversarial trickery"*, and *"keep tasks unambiguous"* as explicit objectives in this unified search space lets the meta agent optimize difficulty, diversity, and alignment constraints simultaneously. Because each generated anomaly carries its provenance and (optionally) a machine-checkable proof, the resulting benchmark remains auditable even as it grows without bound. In short, a meta-agent layer transforms ATAD from a fixed three-role protocol into a self-refining ecosystem where agents, workflows, and evaluation criteria co-evolve alongside the LLMs they test.