# OpenReview forum: "From Static Benchmarks to Dynamic Protocol: Agent-Centric Text Anomaly Detection for Evaluating LLM Reasoning"
_NeurIPS.cc/2025/Datasets_and_Benchmarks_Track — Submitted to NeurIPS 2025 Datasets and Benchmarks Track_

### Official Review · Reviewer_DYiT · 2025-06-26

**Rating:** 4
**Confidence:** 3

**Summary:**

The paper presents a dynamic benchmarking protocol for evaluating large language models called ATAD (Agent-Centric Text Anomaly Detection). Unlike traditional static benchmarks, ATAD uses a system of three autonomous agents: a teacher, an orchestrator, and a student. The protocol generates, validates, and solves text anomaly problems, which evolve in difficulty as models improve. This approach helps identify subtle reasoning flaws that are often missed by static benchmarks. The paper introduces text anomaly detection tasks, including tasks related to logical consistency, coherence, and ambiguity, and demonstrates that ATAD effectively surfaces reasoning weaknesses in LLMs.

**Dataset Code Accessibility:**

Yes

**Ethical Considerations:**

No, there are no or only very minor ethics concerns

**Final Justification:**

I have decided to accept it and maintain my score.

**Limitations Weaknesses:**

1. Although the authors suggest that the use of LLM as a more dynamic evaluator is very novel and practical, there may still be some bad cases. Did the authors do a manual check to find out if some bad cases passed the proposed validation mechanism and analyze why?
2. The dynamic nature of the ATAD protocol, which involves iterative problem generation and validation, could lead to increased computational demands. Each round of difficulty scaling requires multiple interactions between the Teacher, Orchestrator, and Student agents, which may not be efficient for large-scale benchmarking, especially with many models or a diverse set of tasks. This could limit the feasibility of scaling ATAD for extremely large evaluations or for continuous benchmarking with rapidly evolving models.

**Strengths Contributions:**

1. ATAD introduces a novel, agent-centric framework for benchmarking LLMs, moving beyond static datasets to dynamically generated problems that evolve in complexity.
2. The dynamic difficulty scaling ensures that the benchmark adapts to increasingly capable models, providing ongoing relevance in the face of rapid advancements in LLM technology.
3. The benchmark spans various types of text anomalies, covering multiple reasoning skills like contextual coherence, contradiction detection, and referential clarity, offering a comprehensive evaluation of LLM capabilities.
4. The paper presents thorough empirical results, showing that ATAD identifies weaknesses in models that are not detectable by conventional benchmarks. It also includes comprehensive data and code, supporting reproducibility.

---

> ### Author Rebuttal · Authors · 2025-07-30
>
> Thank you for your thoughtful comments. We address your first point regarding validation quality below.
>
> ### **1. On the Observed Quality of Validated Problems**
>
> Thank you for raising this important point. We appreciate the opportunity to clarify how problem quality is managed and observed in our benchmark.
>
> While we did not include a formal manual annotation phase in this version, we closely examined the examples presented in **Supplement Section E (Figures 1 and 2)**. These illustrate how the Orchestrator contributes meaningfully to the refinement process—identifying unclear or insufficiently distinctive anomalies and guiding the Teacher to improve them in a targeted and pedagogically helpful manner. For instance:
>
> - **Case 1: Refining a weak anomaly.**
>     - **Situation:** In one case (**Supplement, Figure 1**), a problem about Kant's philosophy was initially rejected.
>     - **Orchestrator's Feedback:** The Orchestrator noted that the anomaly was not distinctly identifiable and provided specific guidance:
>
>         > "...not 'semantically inconsistent or off-topic enough to be distinctly identifiable'..." and suggested "...introducing a sentence that diverges more distinctly from the topic."
>         >
>     - **Outcome:** Following this guidance, the Teacher created a **revised, approved problem** on existentialism with a much clearer, factually incorrect anomaly. This shows the Orchestrator's role in actively improving problem quality.
> - **Case 2: Rejecting an ambiguous problem.**
>     - **Situation:** In another example (**Supplement, Figure 2**), a multiple-choice question was **rejected for ambiguity** because options like *'hyperrealism'* and *'literary impressionism'* were debatable choices rather than clearly incorrect ones.
>     - **Orchestrator's Feedback:** This ambiguity was noted to:
>
>         > "...undermine both the fairness and the intended difficulty of the problem."
>         >
>     - **Validation & Impact:** Crucially, the Orchestrator's judgment was empirically validated; the ambiguity led **two out of three strong LLMs to answer incorrectly**. Without intervention, this would have created a significant **fairness issue** by including a flawed question in the benchmark. This demonstrates the Orchestrator's critical ability to filter out such items.
>
> As these cases demonstrate, the Orchestrator's multi-stage validation is central to ensuring the quality and fairness of our benchmark. This rigorous process results in finalized items of consistently high quality, a finding further supported by the quantitative evaluations in the main paper (Table 3). For full transparency, all related code and data are open-sourced. As such, our analysis focused on qualitatively reviewing these rejected "bad cases" to validate the Orchestrator's effectiveness, as you suggested. This process confirmed its crucial role in preventing flawed items from passing into the final benchmark, and we will add a summary of this analysis to the final version. Thank you again for this thoughtful point.
>
>
> ### **2. On Scalability and Future Directions**
>
> Thank you for raising this thoughtful concern regarding computational scalability. We agree that the dynamic, multi-agent nature of ATAD introduces computational considerations, particularly for large-scale or continuous evaluations. While the protocol's design allows for parallelization across samples, we view this challenge not as a limitation, but as the next critical research frontier that our new paradigm opens up.
>
> To this end, we are actively exploring several directions for creating more efficient, modular, and resource-aware versions of ATAD, as outlined in our future work (**Supplement, Section H**). Specifically:
>
> - **Modular Sub-Agent Architecture:**
>
>     We envision decomposing the single Orchestrator into a team of specialized, lightweight sub-agents (e.g., a *StructureValidator*, *CoherenceChecker*, *DifficultyCalibrator*). This modularity allows for replacing a single, costly monolithic model with smaller, more efficient LLMs that are conditionally invoked only when needed, improving both efficiency and interpretability.
>
> - **Task-Aware Orchestration:**
>
>     Not all tasks demand equally complex validation. For instance, sentence ordering tasks (T2) can be validated with lightweight checks, while logical contradiction tasks (T6) benefit from deeper semantic analysis. We plan to introduce task-specific validation strategies, enabling the Orchestrator to apply only the necessary evaluation components for each task type, further pruning redundant computational steps.
>
> - **Advanced Resource-Aware Frameworks:**
>
>     As detailed in our future work (**Supplement, Section H**), we plan to formalize these ideas into a more advanced framework.
>
>     - First, we are exploring casting the ATAD loop as a **cooperative game**. This would allow us to use game-theoretic tools like Shapley values to quantify the marginal contribution of each agent's action to benchmark quality. By assigning credit this way, we can dynamically allocate computational resources only to the most valuable validation steps and prune redundant checks, directly enhancing efficiency.
>     - Furthermore, we envision leveraging **meta-agents** that can program and discover superior, more efficient sub-agents in code (Supplement, Section H.2). This approach aims to create specialized validators from smaller models that can rival larger ones at a fraction of the cost, ultimately creating a **self-refining ecosystem** where agents and workflows co-evolve.
>
> Ultimately, our primary contribution with this paper is to initiate a paradigm shift: from **static benchmarks** to **dynamic protocols**. We believe the questions you've raised regarding scalability and modularity are the crucial next step in this new direction. Our goal is for ATAD to serve as a starting point for a broader academic conversation on the next generation of LLM evaluation—a conversation centered on creating the scalable, adaptive, and sub-agentic frameworks that will be necessary to keep pace with the rapid evolution of AI.
>
> Thank you again for highlighting this important consideration.

---

### Official Review · Reviewer_rR1o · 2025-07-01

**Rating:** 4
**Confidence:** 4

**Summary:**

Overused static evaluation datasets/leaderboards may not provide a good estimate of the improvements in the text reasoning abilities of LLMs. This paper proposes a framework to construct dynamically evolving evaluation datasets to address this issue. The proposed framework has a model that generates problems, an orchestrator model that verifies the validity of these problems, and a model that solves these problems. The orchestrator model prompts the problem generation model to generate challenging variants when a problem is solved by the solver model. This way, the evaluation benchmark can be scaled in difficulty by using more capable solver models.

Specifically, this paper focuses on the task of text anomaly detection and generates evaluation samples that focus on seven types of anomalies and six different domains. Experiments were conducted by using four LLMs to generate the evaluation benchmark and testing several LLMs on the generated benchmark.

**Dataset Code Accessibility:**

Yes

**Dataset Code Comments:**

The dataset and code are provided through Huggingface.
https://huggingface.co/datasets/LGAI-DILab/ATAD

**Ethical Considerations:**

No, there are no or only very minor ethics concerns

**Final Justification:**

The rebuttal provided by the authors has addressed the concerns I had. I encourage the authors to modify the paper so that they position their work clearly with respect to the existing literature.

**Limitations Weaknesses:**

**Evaluation benchmark**
* The main drawback of this work is that it focuses only on generating (hard) samples with anomalies, but not (hard) samples without anomalies. When focusing on a binary detection task, i.e., a 2-class problem, the evaluation benchmark that only has one class data is half complete and doesn't give a full picture about the model capabilities. For example, a model that always says there is some anomaly will get 100% accuracy on the generated benchmarks but it will be a useless model.
* The benchmark is generated and verified by an LLM. While modern LLMs are good at instruction following, given the complicated and constrained nature of this generation/verification task, there could still be errors in this synthetic benchmark. There is no analysis done to estimate the extent of errors in the generated dataset.
* The generated benchmark is not compared with existing text anomaly detection benchmarks in terms of difficulty and problem quality. So, the paper does not provide a clear picture of where the proposed benchmark stands with respect to existing benchmarks.

**Results**
* The evaluation dataset used for Table. 3 is generated using GPT-4o according to line 283. I don’t understand why the problem quality metrics vary across the four rows in this table when the evaluation dataset is the same.

**Presentation**
* The paper doesn't discuss works related to dynamic benchmarking and text anomaly detection in the main paper. This discussion is moved completely to the supplementary material. This makes this effectively a "longer than 9-page paper" or "a paper that doesn't position itself well with respect to the existing literature" (within the main paper).

**Strengths Contributions:**

* Overall, the paper was written well and it was easy to follow.

* The proposed dynamic evaluation framework enables the benchmark to scale with the emerging capabilities of language models thereby providing a sustainable evaluation over time.

* Having an orchestrator that checks the generated samples based on various criteria helps in mitigating the issue of adversarial samples. Table 3 shows that having an Orchestrator increases the quality of generated samples.

* Table 2 clearly shows that the difficulty of the samples increases when following the proposed generation process.



* The experimental results show that evaluation benchmark created using one LLM generalizes well to other LLMs in terms of sample difficulty.

---

> ### Author Rebuttal · Authors · 2025-07-29
>
> We sincerely thank the reviewer for their detailed feedback and for recognizing several key strengths of our work. We are grateful for the positive comments on the paper's clarity, the contribution of our dynamic framework to sustainable evaluation, the vital role of the Orchestrator in ensuring sample quality (as shown in Table 3), and the clear evidence of difficulty scaling (Table 2). We appreciate the opportunity to clarify the points raised, and offer our detailed responses below.
>
> ### **1. Clarifying the Scope and Evaluation Paradigm of Our Benchmark**
>
> We would like to clarify a key point regarding the concern about class balance: our benchmark is not designed as a simple binary classification task (i.e., “anomalous” vs. “normal” labels at the sample level). Instead, it is structured as a **localized anomaly identification task**, where each input includes both normal and anomalous elements. This design inherently avoids the risk of degenerate strategies.
>
> - **Localization Tasks (T1, T3, T4, T5, T6, T7):**
>
>     Specifically, in tasks such as T1 (Sentence Context Anomaly), T5 (Referential Ambiguity), T6 (Logical Contradiction), and T7 (Tone/Style Violation), the input consists of a short paragraph (5–6 sentences), and the model must identify **which specific sentence** is anomalous.
>
>     Similarly, tasks T3 (Blank-based Choice Anomaly) and T4 (Bridge Sentence Evaluation) require the model to select the most inappropriate or incoherent choice from a list of five candidates.
>
>     This formulation inherently avoids the risk of degenerate strategies, such as always predicting the presence of an anomaly, since every sample requires **fine-grained selection** rather than binary prediction.
>
> - **Balanced Binary Task (T2):**
>
>     In the case of T2 (Paragraph Order Consistency), which is framed as a binary decision (i.e., whether a paragraph is logically coherent), our dataset includes a **balanced mixture of coherent and incoherent examples**, selected through randomized generation procedures. This ensures that models cannot trivially perform well by predicting one class.
>
>
> We hope this clarification helps situate our benchmark more accurately. In future revisions, we will emphasize this evaluation format more explicitly to avoid such confusion.
>
>
> ### **2. On Benchmark Validity and LLM-Based Verification**
>
> Thank you for this important point regarding benchmark validity. We agree that ensuring the quality of automatically generated benchmarks is essential. Our protocol incorporates several key mechanisms to address this challenge:
>
> - **Principled Generation with Structured Prompts**
>
>     First, the **Teacher model** in our protocol does not generate anomalies heuristically. Instead, each anomaly is designed to target a specific type of reasoning (e.g., contextual inference, referential clarity, logical contradiction), and is guided by structured prompt templates grounded in **public exam styles** (e.g., GRE, GMAT). These templates define both the rhetorical framing and the level of subtlety required, ensuring that generation is systematic, diverse, and difficulty-controlled. (Detailed in **Supplement Section D.1**. )
>
> - **Rigorous, Multi-Stage Validation by the Orchestrator**
>
>     Second, the **Orchestrator agent** performs structured, context-aware validation at two critical stages:
>
>     (1) After base problem generation, to ensure the problem is well-formed, fair, and solvable.
>
>     (2) After difficulty escalation, to verify that harder version preserves clarity while increasing challenge.
>
>     This multi-turn validation loop—where the Orchestrator observes the Student’s performance and uses that signal to guide and evaluate Teacher revisions—enables more **nuanced, phase-specific judgment** than a single-pass filter. We detail the orchestrator’s criteria and role in **Supplement Section D.2**.
>
>
> **Evidence of Quality Control**
>
> We provide both quantitative and qualitative evidence to support the effectiveness of this process:
>
> - **Quantitative Evidence:** In the main paper (**Table 3**), we show that the Orchestrator's involvement significantly improves sample quality across validity, coherence, and fairness metrics, as judged by multiple independent LLMs.
> - **Qualitative Evidence:** The supplement (**Section E**) provides case studies illustrating how the Orchestrator filters vague or ambiguous problems (**Supplement, Figure 2**) and guides the refinement of weak ones into clear, pedagogically effective benchmark items (**Supplement, Figure 1**).
>
> While we acknowledge that no generation pipeline is immune to imperfections, our protocol’s design—particularly the combination of **task-specific generation**, **Orchestrator-guided multi-stage validation**, and **iterative refinement**—aims to minimize such risks and promote high-quality, reliable samples. We will clarify this robust process more explicitly in the final version.
>
> ### **3. Positioning Against Prior Text Anomaly Detection Benchmarks**
>
> Thank you for this valuable comment. We agree that clearer positioning with respect to prior work is essential. To that end, we provide both a qualitative comparison to clarify the fundamental differences in approach and an explanation regarding the challenges of a direct quantitative comparison.
>
> | Feature | **ATAD (Ours)** | **CoheSentia** | **DECOR** | **Disco-Bench**  |
> | --- | --- | --- | --- | --- |
> | **Paradigm** | **Dynamic Protocol**  | Static Dataset  | Static Dataset  | Static Benchmark Suite  |
> | **Difficulty Scaling** | **Adaptive & Co-evolutionary** (scales with model performance)  | Fixed / Static  | Fixed / Static  | Fixed / Static  |
> | **Quality Control** | **Automated Validation** (Orchestrator for clarity, fairness)  | Human Annotation  | Expert Labeling  | Curated from existing tasks  |
> | **Primary Goal** | Fine-grained **Reasoning Diagnostics**  | Holistic Coherence **Scoring**  | L2 Writing **Incoherence Analysis** (Detect, Explain, Rewrite)  | Broad **Discourse Phenomena** Evaluation  |
> | **Scalability** | **High** (automated generation)  | **Low** (human-intensive)  | **Low** (expert-intensive)  | **Low** (Non-extensible)  |
>
> As the table highlights, ATAD's core novelty lies in its dynamic, agent-centric protocol. This approach is designed specifically to address the **"Clarity-Difficulty Trade-off"** inherent in text anomaly benchmarks. Unlike static datasets that rely on costly human curation and often struggle to produce problems that are both complex and unambiguous, ATAD's protocol uses an automated, regulated process. The Orchestrator agent allows difficulty to increase incrementally while simultaneously ensuring problems remain fair and clear. This enables ATAD to co-evolve with frontier models—a key feature that static benchmarks cannot offer.
>
> **On the Challenge of Direct Quantitative Comparison**
>
> We acknowledge the desire for a direct quantitative performance comparison. However, a fair, apples-to-apples comparison is methodologically challenging due to fundamental differences in task structures and objectives:
>
> - **Divergent Task Formats:** Our tasks are primarily structured as **index selection** or **binary classification**. In contrast, CoheSentia requires models to output coherence **scores**, and DECOR includes complex sub-tasks like **explaining the cause** of an error and **rewriting the sentence**. A single accuracy metric cannot capture performance across these varied formats.
> - **Different Evaluation Goals:** ATAD is designed to diagnose specific reasoning failures, whereas other benchmarks may aim to evaluate coherence scoring or analyze errors in non-native writing. Comparing raw scores across these different goals would not provide a meaningful insight into which benchmark is "more difficult" or "higher quality."
>
> Thank you for pushing us to clarify this important point. Your feedback has helped us crystallize our positioning and identify a clear path for future work, such as exploring ways to bridge the structural gaps for more standardized comparative evaluation. We will ensure this rationale is made explicit in the final version of the paper.
>
>
> ### **4. Clarifying Table 3: Cross-Model Evaluation of Problem Quality**
>
> Thank you for pointing this out. You are correct that the evaluation dataset in Table 3 was generated using GPT-4o. The reason the quality metrics vary across rows is that we asked *different LLMs*—including GPT-4o, Gemini-2.0-Flash, Claude-3.5-Sonnet, and LLaMA-3.3-70B—to serve as **reviewers** of the same benchmark.
>
> Each model independently evaluated the samples on *validity, coherence,* and *fairness*, based on its own reasoning capabilities and internal standards. We designed this setup intentionally to assess whether improvements introduced by the Orchestrator generalize across diverse evaluators, rather than relying on a single model’s judgment.
>
> We appreciate that this could be clearer in the current version and will revise the description in Table 3 and accompanying text to better convey this setup.
>
> ### **5. Related Work Placement in the Supplement**
>
> This is a very fair point, and we appreciate you raising it. In preparing this paper, we felt a thorough literature review was essential to properly contextualize our contribution. Our work intersects with dynamic benchmarking, text anomaly detection, and agent-based evaluation, and a comprehensive discussion grew to nearly three pages. (**Supplement, Section G**)
>
> Rather than presenting a heavily truncated version in the main paper that might omit crucial context, we chose to provide the full, in-depth review in the supplement. However, we agree that this created a gap. For the final version, we will integrate a focused summary of the most relevant works into the main text, ensuring our work is well-positioned while retaining the comprehensive review for interested readers.

---

> > ### Comment · Reviewer_8aYk · 2025-08-01
> >
> > Dear authors,
> > I was reading this answer, and for the "Difficulty Scaling" the table had "Fixed / Static" for all methods
> >
> > Is it a mistake, or could you please explain what it means?

---

> > > ### Author Response · Authors · 2025-08-02
> > >
> > > ### **Response to Reviewer 8aYk (Difficulty Scaling Clarification)**
> > >
> > > Thank you so much for carefully reviewing our discussion and for raising this important clarification. We truly appreciate your engagement—it’s a rare and valuable opportunity for us as authors to refine our claims, and your comment about “Difficulty Scaling” helped us realize that our original table labeling may have introduced ambiguity. So we are especially grateful that you brought it up during the discussion phase—it gave us a chance to clarify a central aspect of our benchmark design.
> > >
> > > ---
> > >
> > > ### **Clarifying "Difficulty Scaling" Across Benchmarks**
> > >
> > > You are absolutely right to question why existing benchmarks were marked as “Fixed / Static” under “Difficulty Scaling.” We’d like to clarify the distinction we intended to draw:
> > >
> > > | Feature | ATAD (Ours) | CoheSentia | DECOR | Disco-Bench |
> > > | --- | --- | --- | --- | --- |
> > > | Difficulty Scaling | **Protocol-level Adaptive** (task difficulty evolves based on student model behavior) | Sample-level Variance (some easy/hard examples) | Multiple levels, pre-defined by human experts | Mixed difficulty, from existing tasks |
> > >
> > > While other benchmarks may include samples of varying difficulty, or even group items into easy/hard tiers, they do not *adaptively generate* harder problems in response to model behavior. Once such datasets are created, their **difficulty distribution becomes fixed** and does not change across evaluation runs or model improvements. Hence, we used the term "Fixed / Static" to indicate the **non-evolving nature** of these datasets.
> > >
> > > In contrast, ATAD is not a static dataset but a **dynamic protocol**—its benchmark can be regenerated as models evolve, with the protocol automatically adjusting the difficulty through agent interactions.
> > >
> > > That is, ATAD’s difficulty scaling is **protocol-driven**:
> > >
> > > - A Teacher–Student loop iteratively escalates the challenge whenever the Student succeeds.
> > > - The Orchestrator ensures that increasing difficulty does not compromise fairness or clarity.
> > >
> > > This dynamic allows the benchmark to co-evolve with model capabilities—a feature unavailable in static datasets.
> > >
> > > We fully agree that our original table could have been clearer in differentiating between sample-level difficulty variance and protocol-level adaptive scaling, and we will revise this in the final version accordingly (e.g., relabeling CoheSentia and DECOR as “Sample-level Variance” and ours as “Adaptive Protocol-level Scaling”).
> > >
> > > ---
> > >
> > > ### **Why This Clarification Matters**
> > >
> > > As you insightfully pointed out, the phrase “Fixed / Static” may mislead readers if interpreted as referring only to the presence or absence of easy/hard samples. But ATAD is not just a dataset—it is a **protocol** that adaptively regulates, escalates, and finalizes problem difficulty through agent interaction.
> > >
> > > Thank you again for prompting us to be more precise in expressing this distinction. We find this kind of dialogue deeply valuable—not only for refining our current paper but also for clarifying the broader framing of dynamic benchmarks for future work. We’d be happy to continue this discussion if you have further thoughts.

---

> ### Comment · Reviewer_rR1o · 2025-08-04
> **Thank you for the rebuttal**
>
> I thank the authors for their thorough rebuttal. Most of my concerns were addressed and I am now leaning towards acceptance.

---

> > ### Author Response · Authors · 2025-08-05
> >
> > Thank you very much for your thoughtful review. We truly appreciate your engagement, and we’re glad to hear that our response addressed most of your concerns. Your feedback has been valuable in refining our work, and we will reflect the improvements in the final version.

---

### Official Review · Reviewer_Z8Fg · 2025-07-02

**Rating:** 5
**Confidence:** 3

**Summary:**

This paper proposes a novel benchmarking framework that moves beyond static datasets by introducing a multi-agent protocol for the dynamic generation and validation of reasoning-intensive tasks for LLMs. The system includes a Teacher agent generating text anomaly problems, a Student agent solving them, and an Orchestrator validating clarity, difficulty, and fairness. Tasks become progressively harder as the Student succeeds, enabling adaptive, instance-level difficulty scaling.

**Additional Feedback:**

* please, add a related work section. recently, there has been several studies showing temporal misalignment of static data and evaluations over time.

**Dataset Code Accessibility:**

Yes

**Ethical Considerations:**

No, there are no or only very minor ethics concerns

**Final Justification:**

I'll keep my review score as the authors addressed my comments and agreed to include them in the paper.

**Limitations Weaknesses:**

* focus on English
* the protocol is quite complex, which might make the application of this idea to other cases not straightforward
* related work section is missing

**Strengths Contributions:**

* interesting and relevant paper to the research community, which addresses an important challenge in model benchmarking, i.e., their being static
* novel methodology to introduce agents for dynamic benchmarking
* analysis on seven task types of text anomalies

---

> ### Author Rebuttal · Authors · 2025-07-29
>
> Thank you for your positive and encouraging review. We appreciate your recognition of the importance of dynamic benchmarking and the novelty of our agent-based protocol. Below, we address your comments and suggestions in turn.
>
> ### **1. On the focus on English**
>
> We acknowledge that the current implementation of ATAD is focused on English. This design choice was made to align with existing evaluation protocols and widely used reasoning benchmarks (e.g., GRE, GMAT). That said, we believe the **agent-centric framework is language-agnostic by design**—both the generation (Teacher) and validation (Orchestrator) rely on LLMs that support multilingual capabilities. Extending ATAD to other languages is an exciting direction we are actively exploring and will briefly note as future work in the revised manuscript.
>
> ### **2. On complexity of the protocol**
>
> We appreciate this point. The multi-agent design inevitably introduces some complexity; however, we suggest that this complexity is functional and modular. Each agent plays a clearly defined and generalizable role that can be adapted to various tasks beyond anomaly detection:
>
> - **The Teacher** is responsible for generating new problems according to ***the specifications of a given task type***. Its role is to create content that fits a predefined format and difficulty, not just anomalies.
> - **The Student**'s role is to perform ***the defined task***, such as selecting the correct index, making a binary choice, or generating a response. Its success or failure provides the core signal for the adaptive difficulty scaling, making it the evaluation target for that specific task.
> - **The Orchestrator** functions as a quality-control module, ensuring that each generated problem is well-defined, coherent, and fair according to ***the rules of the specific task***. It validates task adherence, not just anomaly clarity.
>
> Moreover, the protocol has been implemented as a ***reusable*** framework for other use cases ***by simply replacing the task definition***, and we will open-source the full codebase with detailed examples to facilitate adoption. In the revised paper, we will include a diagram to better illustrate the workflow and clarify how users can adapt or simplify it for their own settings.
>
>
> ### **3. On missing related work section**
>
> Thank you for this suggestion. We already include a discussion of related work on static benchmark limitations and adaptive evaluation in the Supplement (Section G), and will integrate a concise Related Work section into the main paper to improve accessibility.
>
> We thank you again for your helpful feedback and look forward to integrating these improvements in the final version.

---

> > ### Comment · Reviewer_Z8Fg · 2025-08-05
> >
> > Thanks to the authors for their rebuttal and their agreement in including these comments in the paper. I confirm my score for the paper.

---

> > > ### Author Response · Authors · 2025-08-06
> > >
> > > We truly appreciate your thoughtful comments. They’ve helped us improve the work, and we look forward to reflecting them in the final version.

---

### Official Review · Reviewer_8aYk · 2025-07-03

**Rating:** 4
**Confidence:** 3

**Summary:**

This paper proposes a benchmark for LLMs that automatically updates their difficulty level. The benchmark is based on text anomaly detection. Namely, a sentence that does not suit the sentence (in terms of flow, topic, logic, or so) is inserted into a given text, and the tested model has to find that sentence. While relying on a given LLM (“teacher”) to generate such challenges, an orchestrator model judges that the challenge is still “fair”. In that way, the teacher can produce anomaly detection challenges of increasing complexity, generating a challenge of a suitable difficulty to the tested model.

**Additional Feedback:**

a1. Do you have any comments on the ability to fine-tune models using this metric, in a manner somewhat similar to GANs?
a2. Have you considered other practical use cases of text anomaly detection?

**Dataset Code Accessibility:**

Yes

**Dataset Code Comments:**

Dataset URL: https://huggingface.co/datasets/LGAI-DILab/ATAD
Code URL: https://github.com/seungdongy/atad

**Ethical Considerations:**

No, there are no or only very minor ethics concerns

**Final Justification:**

I like some aspects of the papers, and I tend to recommend acceptance.
Yet, I am not knowledgeable enough about some of the related topics to have a stronger opinion on this work.

**Limitations Weaknesses:**

1. Crucial parts of the methods are not described in the manuscript. In which exact manner are the anomalies generated? How does the orchestrator decide if a challenge is “fair”?
2. Yet, some other parts of the text are repetitive (e.g., Sec 3.1)
3.It is also not clear to what extent the performance in this benchmark correlates with other properties users might care about.
4. Section 3.2 can also be better explained.

**Strengths Contributions:**

1. The benchmark is nicely motivated in the introduction.
2. The idea of adjusting the task difficulty using the described method is interesting and, to the best of my knowledge, novel.
3. The authors evaluate their benchmark across a few models, and also do an experiment to simulate future relevance.

---

> ### Author Rebuttal · Authors · 2025-07-29
>
> Thank you for your thoughtful and constructive review. We appreciate the positive feedback on our motivation and novelty. Below, we address your concerns and will incorporate these clarifications into our revision.
>
> ### **1. Clarifying the Core Mechanism: Anomaly Generation & Orchestrator Fairness**
>
> This is a crucial point, and we agree that these mechanisms deserve a more detailed explanation. Our system operates on a structured, multi-stage protocol to ensure robust and principled evaluation.
>
> **1-A. How Anomalies are Generated (by the Teacher): Task-Specific & Difficulty-Controlled**
>
> The anomaly generation is a principled process guided by structured prompts and task designs, not random insertion.
>
> - **Targeted Reasoning Skills:** Each of our 7 tasks is designed to probe a distinct reasoning skill (e.g., contextual inference, logical contradiction), as detailed in **Supplement Table 3**.
> - **Exam-Style Grounding:** Generation is guided by public exam formats (e.g., GRE, GMAT), which are paired with appropriate domains (e.g., philosophy for GRE-style semantic deviation) to ensure realism and pedagogical coherence (**Supplement Tables 2 & 3, `tasks_config.py`**).
> - **Difficulty-Scaling via Prompts:** We control difficulty using structured prompt templates (**`prompt_templates.py`**) that instruct the Teacher on the type and subtlety of the anomaly required. For a concrete example of a task-specific prompt, please see **Supplement Section D.1**.
>
> **1-B. How Fairness is Validated (by the Orchestrator): Phase-Specific & Context-Aware**
>
> The Orchestrator acts as a dynamic quality-control agent that makes nuanced, context-aware judgments at each phase. It evaluates problems at two critical phases with rich context:
>
> - **Phase 1: Base Problem Validation:** It first ensures the initial problem is well-formed, logically coherent, and has a clear, unambiguous solution according to our validation criteria (**Validity, Type Adherence, Coherence, Fairness**).
> - **Phase 2: Difficulty Escalation Validation:** After a Student agent successfully solves a problem, the **Teacher is prompted to generate a more challenging variant**. The Orchestrator then assesses this new version, checking not only for fairness but also ensuring the difficulty has meaningfully increased without sacrificing clarity.
>
> This multi-turn process allows the Orchestrator to make nuanced judgments based on the student's performance and the problem's evolution. The prompts guiding these validation steps are detailed in **Supplement Section D.2**. The effectiveness of this process is demonstrated in **Supplement Figures 1 & 2**, which show the Orchestrator rejecting ambiguous problems and guiding the refinement of weak ones into high-quality challenges.
>
> We will revise Sections 2.2 and 3.1 to make this two-stage validation flow and the structured nature of anomaly generation explicit in the main paper.
>
>
> ### **2. Paper Structure, Clarity, and Benchmark Relevance**
>
> - **On Paper Structure and Clarity (Sec 3.1 & 3.2):** We appreciate the feedback. We will **condense the repetitive parts of Sec 3.1** as you suggested. For **Sec 3.2, we will restructure it for better clarity**. To do this, we will introduce a summary table in the main paper—consolidating key details from **Supplement Tables 2 & 3**—that outlines each task’s **targeted reasoning type, input/output structure, and challenge factors**. This will make our task design more immediately accessible to readers.
> - **On Benchmark Relevance:** Regarding the performance relevance of our benchmark, ATAD is designed primarily as a **reasoning diagnostic tool**, not to replace all downstream task evaluations, but to surface subtle failure modes that traditional benchmarks can miss. Its tasks directly correlate with practical applications:
>     - **T5 (Referential Ambiguity):** Crucial for validating summarization and preventing hallucination.
>     - **T6 (Logical Contradiction):** Essential for fact-checking and safety auditing in QA systems.
>     - **T7 (Tone/Style Violation):** Relevant for ensuring stylistic consistency in generated text.
>
> We will expand this discussion in the revised paper to make the connection to real-world applications clearer.
>
>
> ### **3. Additional Feedback**
>
> - **a1. On Fine-tuning (GAN-like):**
>
>     Thank you for this excellent question that highlights our protocol's long-term value.
>
>     The data generated by ATAD can indeed be used for fine-tuning. However, our framework leverages this to enable continuous growth, not to reach a static endpoint. A model fine-tuned on our data becomes a more capable Student agent. This, in turn, compels the Teacher to generate more subtle and complex anomalies to challenge it, creating a co-evolutionary cycle where both the model's ability and the benchmark's difficulty evolve together.
>
>     This competitive dynamic is where the parallel to GANs is most apparent. In our protocol, this process is guided by the Orchestrator agent, which acts as a crucial quality regulator. It ensures the Teacher generates genuinely harder reasoning challenges, rather than simply ambiguous or unfair "adversarial examples"—a known risk in GANs. This regulation is key to our framework's main objective: to create a benchmark that sustainably and fairly evolves, constantly pushing the frontier of model capabilities.
>
> - **a2. On Practical Use Cases:**
>
>     We see direct applications of text anomaly detection in safety auditing (detecting logical flaws), content validation (checking summaries for coherence), and instructional feedback (diagnosing reasoning gaps in educational tools) , due to the ability to verify reasoning or logical flaws. ATAD provides a structured taxonomy for building these practical diagnostic modules.
>
> We sincerely appreciate your insightful feedback and are confident that these revisions will significantly improve the paper.

---

> > ### Comment · Reviewer_8aYk · 2025-08-01
> >
> > I would like to thank the authors for their answers.
> > Yet, I would still like to have further clarification about two issues:
> >
> >
> > 1. ***"Phase 1: Base Problem Validation: It first ensures the initial problem is well-formed, logically coherent, and has a clear, unambiguous solution according to our validation criteria (Validity, Type Adherence, Coherence, Fairness)."***
> >
> > It is not clear to me yet how the suggested process ensures that. An LLM judge can, of course, encourage that, but the extent to which this is ensured is limited by the quality of the judge.
> >
> > 2. ***"However, our framework leverages this to enable continuous growth, not to reach a static endpoint. A model fine-tuned on our data becomes a more capable Student agent."***
> >
> > This sounds somewhat speculative. While I believe this may indeed improve weaker models to some extent. I am not sure if this method will lead to an ever-increasing capability via self-training.
> > In any case, I acknowledge that this exciting possibility is not part of the claims in the original paper :-)
> >
> > More relevant to the original claims is the discussion of the possible saturation of the benchmark. I think that claiming the benchmark will never saturate might be too strong and require further evidence (see also 1 in this comment).

---

> > > ### Author Response · Authors · 2025-08-02
> > >
> > > ### **Response to Comment 1 – On the Quality of Orchestrator Judgment**
> > >
> > > We sincerely thank the reviewer for the thoughtful follow-up. Your observation—that **the reliability of LLM-based judgment (in our Orchestrator)**—is both valid and insightful. This is an area we have thought deeply about, and your comment helps sharpen the presentation of our current protocol as well as the future path forward.
> > >
> > > ---
> > >
> > > ### **(a) Clarifying the current design**
> > >
> > > First, we want to clarify how the current Orchestrator works beyond a naïve “LLM-as-judge” setup:
> > >
> > > - The Orchestrator is instantiated as an **independent role** from the Teacher and Student agents.
> > > - It operates under **task-specific, rubric-based prompts** assessing Validity, Type Adherence, Coherence, and Fairness.
> > > - In the **base validation phase**, the Orchestrator checks whether the problem is logically well-formed, unambiguous, and pedagogically coherent.
> > >
> > > **Importantly—and this directly addresses your point—the Orchestrator does receive the Teacher’s reference answer, but it does not merely verify the answer in isolation.** Rather, it evaluates the problem holistically—checking whether the formulation is logically sound, unambiguous, and pedagogically coherent, including whether the reference answer appropriately reflects the intended reasoning. This helps avoid circular biases or hallucinated alignment that might arise from overly fixating on surface-level correctness.
> > >
> > > **While this setup does not eliminate all the limitations of LLM-based judgment, it substantially reduces circular alignment risks by** decoupling generation and validation, and by grounding evaluation in task-specific reasoning criteria rather than surface answer agreement.
> > >
> > > ---
> > >
> > > ### **(b) Empirical Validation**
> > >
> > > To demonstrate the Orchestrator’s effectiveness, we present both **quantitative** and **qualitative** evidence:
> > >
> > > - **Main Paper Table 3**: Orchestrator validation leads to **significant improvements** across four quality metrics (Validity, Coherence, Fairness, and Approval Rate), as judged by *independent LLMs*:
> > >     - For instance, Approval Rate increases from **38% (w/o Orch.) to 87% (w/ Orch.)**, especially when using a high-capability judge LLM such as GPT-4o.
> > >     - Fairness judgments improve by over **+1.0 point** on a 5-point scale.
> > > - **Supplement Figures 1 & 2**:
> > >     - *Figure 1*: Shows an initially vague T1 item flagged by the Orchestrator and revised into a sharper anomaly targeting scientific conceptual inconsistency.
> > >     - *Figure 2*: Shows a structurally valid T3 item rejected for semantic ambiguity—despite formal adherence—demonstrating the Orchestrator’s context-sensitive discernment.
> > >
> > > These results provide **direct evidence that the Orchestrator improves item quality**, beyond what a one-shot generation would allow.
> > >
> > > ---
> > >
> > > ### **(c) Intuitive parallel & future direction**
> > >
> > > Beyond these results, we believe an intuitive analogy may help clarify the rationale behind our protocol design.
> > > A helpful analogy here is how we interact with LLMs in everyday use:
> > >
> > > > When a response seems vague or misleading, we instinctively offer feedback—"No, try again, but make it clearer or more logical"—and are often rewarded with better outputs.
> > > >
> > >
> > > This kind of iterative refinement mirrors how we often interact with LLMs informally. ATAD draws inspiration from this pattern, formalizing it through role-separated agents and guided revision, enabling quality control that’s both structured and scalable.
> > >
> > > That said, your suggestion—implicitly pointing to the limits of *single-model judgment*—opens up an exciting direction: **can we go further by introducing redundancy or multi-agent arbitration into the Orchestrator itself?**
> > >
> > > We believe this is an excellent idea. In fact, we are currently exploring **dual-agent verification**, where two distinct Orchestrator instances cross-validate the judgment (e.g., one acts as primary judge, and the other as a verifier of the fairness/coherence evaluation). This could mitigate bias, catch blind spots, and further stabilize quality.
> > >
> > > In this sense, your insightful comment doesn’t just identify a limitation—it **suggests a concrete path to a stronger protocol**, and we are grateful for that.

---

> > > ### Author Response · Authors · 2025-08-02
> > >
> > > ### **Response to Comment 2 – On Benchmark Saturation and Model Co-Evolution**
> > >
> > > Thank you for raising the thoughtful point regarding potential benchmark saturation. We would like to clarify that ATAD is fundamentally designed to remain adaptable—capable of evolving alongside increasingly capable agents.
> > >
> > > This co-evolutionary potential is embedded in the interaction among all three roles—**Teacher, Orchestrator, and Student**—each of which can be instantiated with stronger LLMs over time. This structure allows the benchmark to remain challenging and diagnostic, even as models improve.
> > >
> > > ---
> > >
> > > ### **(a) Co-evolution through role-based upgrades**
> > >
> > > The key dynamic is that **as stronger LLMs are adopted across all roles**—Teacher, Orchestrator, and Student—the system naturally generates more difficult and diagnostically precise problems. This enables the benchmark to scale its difficulty in tandem with the evolving model landscape, unlike static datasets.
> > >
> > > In particular:
> > >
> > > - **The Teacher** is more capable of generating subtle, well-targeted anomalies that exploit **emerging blind spots** in advanced Student models.
> > > - **The Orchestrator**, when upgraded, becomes more discerning—**filtering out ambiguous or overly simple problems** and pushing for higher reasoning fidelity.
> > > - **The Student**, when fine-tuned or replaced by stronger base models, reveals **new failure modes** that guide the Teacher’s problem search in future rounds.
> > >
> > > This closed-loop interaction enables the benchmark to **dynamically adjust its difficulty** in tandem with the evolving model landscape—even if any one model saturates, the benchmark itself does not.
> > >
> > > ---
> > >
> > > ### **(b) Empirical Signals of Saturation Resistance**
> > >
> > > In our current implementation, all agents are instantiated with GPT-4o. However, the protocol is **designed to be easily applicable to future models** (e.g., GPT-5 or Claude 5) without requiring additional manual supervision. The **difficulty** of the problems will likely scale **naturally** when stronger agents are used. While we acknowledge that the phrase “never saturate” may be too strong, ATAD is intentionally structured to help mitigate benchmark saturation through its dynamic, agent-centric protocol.
> > >
> > > We are currently working on **difficulty-response patterns across LLM tiers** to document how problem difficulty and failure rates change across models (e.g., GPT-3.5 → GPT-4 → GPT-4o). This ongoing work will help empirically validate the **saturation resistance** of ATAD over time.
> > >
> > > Thank you again for raising this important point. Your comment helped us better articulate that ATAD’s long-term strength lies in **protocol-driven adaptability.** We will reflect this nuance more explicitly in the final version.

---

> ### Comment · Reviewer_8aYk · 2025-08-04
>
> Thank you for the additional explanations. I have no further concerns.

---

> > ### Author Response · Authors · 2025-08-05
> >
> > Thank you for the constructive discussion and careful evaluation. We’re glad our clarifications addressed your points and that you have no further concerns. We’re especially grateful for your contribution to our goal of moving *From Static Benchmarks to Dynamic Protocol*. We will update the manuscript accordingly, and we would be glad to continue the discussion anytime.

---

### Note · Authors · 2025-08-15

We sincerely thank all reviewers for their constructive feedback and discussions throughout the review process. We are encouraged that our clarifications have addressed the main concerns, and we appreciate the reviewers’ recognition of the improvements. We will reflect the reviewers’ helpful suggestions in the final version.

We believe ATAD contributes a novel and timely shift from static datasets to dynamic, agent-centric protocols for evaluating LLM reasoning. We are grateful for the opportunity to refine and present this work, and we look forward to advancing this direction in collaboration with the community.

---

### Decision · Program_Chairs · 2025-09-18

**Decision:**

Reject

**Comment:**

This paper introduces a agent-centric protocol for dynamically evaluating LLM reasoning,  moving beyond the limitations of static benchmarks - to enable scalable, and stable evaluation of ever-evolving LLMs. All the reviewers agree on the novelty of the core idea proposed. At the same time, there are few weaknesses on the clarity and computational demands of the proposed iterative problem generation and validation protocol. The authors' rebuttal address key concerns, providing detailed explanations of the protocol and committing to incorporate the feedback in the final version. Overall, I believe this work presents an interesting new direction for LLM evaluation, and I recommend Acceptance.

===== FINAL UPDATE FROM DB Track PCs ====

The final decision for this paper has been taken by the program chairs after consultation with the SACs. All Senior Area Chairs have ranked papers according to the feedback from the AC during the review process. We decided to leave the original meta-review to reflect the opinion of the AC in light of the initial discussions with reviewers and SAC.